# Probing the dynamic landscape of peptides in molecular assemblies by synergized NMR experiments and MD simulations
Ricky Nencini[1,2], Morgan L. G. Regnier [1], Sofia M. Backlund[1], Efstathia Mantzari[1], Cory D. Dunn[1] & O. H. Samuli Ollila [1,3] ✉

Peptides or proteins containing small biomolecular aggregates, such as micelles, bicelles, droplets and nanodiscs, are pivotal in many fields ranging from structural biology to pharmaceutics. Monitoring dynamics of such systems has been limited by the lack of experimental methods that could directly detect their fast (picosecond to nanosecond) timescale dynamics. Spin relaxation times from NMR experiments are sensitive to such motions, but their interpretation for biomolecular aggregates is not straightforward. Here we show that the dynamic landscape of peptide-containing molecular assemblies can be determined by a synergistic combination of solution state NMR experiments and molecular dynamics (MD) simulations. Solution state NMR experiments are straightforward to implement without an excessive amount of sample, while direct combination of spin relaxation data to MD simulations enables interpretation of dynamic landscapes of peptides and other aggregated molecules. To demonstrate this, we interpret NMR data from transmembrane, peripheral, and tail anchored peptides embedded in micelles. Our results indicate that peptides and detergent molecules do not rotate together as a rigid body, but peptides rotate in a viscous medium composed of detergent micelle. Spin relaxation times also provide indirect information on peptide conformational ensembles. This work gives new perspectives on peptide dynamics in complex biomolecular assemblies.

Nanoscale biomolecular aggregates containing peptides or proteins have applications in a wide range of fields. Micelle, bicelle and nanodisc systems are used to characterize membrane proteins[1,2], high and low-density lipo-proteins (HDL and LDL) are lipid droplets with apolipoproteins attached on their surface[3,4], and nanodiscs stabilized by apolipoprotein mimetic and peptide micelles have potential pharmaceutical applications[5,6]. However, these small and highly dynamic aggregates are inaccessible with most experimental methods that deliver atomistic resolution data of biomolecular systems, such as crystallography or electron microscopy. Nuclear magnetic resonance (NMR) experiments are sensitive to fast (ps to $\mu$s) timescale motions of small aggregates, but interpretation of the data is often not straightforward because (i) reference direction, such as membrane plane that enables direct determination of order parameters, is not well defined in solution state samples, and (ii) methods to interpret conformational

ensembles and dynamics of aggregates of lipid-like disordered molecules from solution state NMR data are not available.

Spin relaxation times, $T_1$, $T_2$ and heteronuclear NOE relaxation (het-NOE), measured with solution state NMR from isotopically labelled $^{15}$N atoms in peptide backbone are often used to determine protein dynamics by exploiting their connection to rotational dynamics of N-H bond vectors via Redfield equations[7,8]. Spin relaxation times from proteins are typically interpreted using Lipari-Szabo formalism or its extensions, where bond vector rotational motions are assumed to be composed of overall motion and one or more independent modes of internal motions, and parameters describing these motions are then solved by fitting to the experimental data[8,9]. However, for peptides or proteins embedded in disordered lipid-like aggregates, it is not clear which kind of rotational modes should be used in these calculations, and if all molecules in aggregates rotate together as a rigid

[1]Institute of Biotechnology, University of Helsinki, Helsinki, Finland. [2]Division of Pharmaceutical Biosciences, Faculty of Pharmacy, University of Helsinki, Helsinki, Finland. [3]VTT Technical Research Centre of Finland, Espoo, Finland. ✉e-mail: samuli.ollila@helsinki.fi

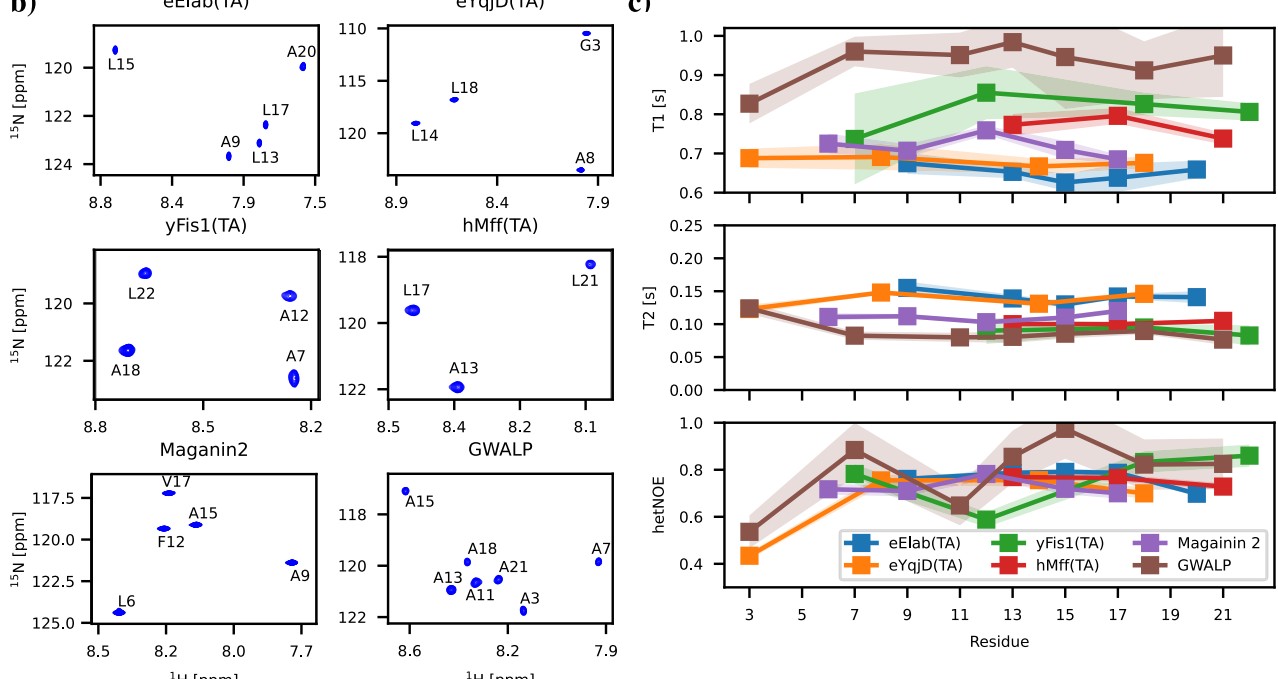

**Fig. 1 | Experimental results for peptides in SDS micelles. a** Amino acid sequences with $^{15}$N labelled residues shown in red for transmembrane GWALP23, peripheral Magainin 2, and mitochondria-directed tail anchor (eElaB(TA), eYqjD(TA), yFis1(TA), and hMff(TA)) peptides. **b** $^{1}$H - $^{15}$N HSQC spectra and (**c**) $T_1$, $T_2$ and hetNOE spin relaxation times measured from the peptides in SDS micelle and sodium-phosphate buffer at 310 K with 850 MHz spectrometer.

object or if peptides rotate independently from other molecules. In addition, the number of molecules in each aggregate and the potential formation of dimers or higher multimers of peptides may affect rotational dynamics. To resolve all these free parameters by fitting would require a large amount of experimental data. On the other hand, complex heterogeneous dynamics of disordered molecules can be resolved by interpreting spin relaxation data directly using molecular dynamics (MD) simulations[10]. However, this approach requires careful exploration of MD simulation models that reproduce experimental spin relaxation times with sufficient accuracy for the interpretation of experimental data[10,11].

Here we present an approach to resolve the dynamic landscape of complexes formed by disordered biomolecules by interpreting spin relaxation data from solution state NMR experiments using MD simulations. This is demonstrated for six different types of peptides embedded in sodium dodecyl sulfate SDS micelles, a common anionic membrane-mimicking environment[2]. For the references, we resolved the dynamic landscape of two widely characterized peptides in the micellar environment: a model transmembrane GWALP peptide[12] and an antimicrobial Magainin 2 peptide that is known to settle in a peripheral orientation parallel to membranes[13]. To further demonstrate the usefulness of our approach, we studied tail-anchored peptides shown to target the mitochondrial outer membrane in yeast [eElaB(TA), eYqjD(TA), yFis1(TA)] or human [hMff(TA)] cells. The mechanism by which these tail-anchored peptides are inserted into the membrane is still poorly understood[14–16]. For these six peptides, we predicted experimental spin relaxation times directly, without any further fitting, from the MD simulation force field based on physical interaction parameters between atoms. Our predictions allowed a detailed interpretation of the dynamic landscapes of peptides and a better understanding of their behaviour in a lipid-like environment. Most importantly,

the approach presented here can be generally applied to characterize dynamic landscapes of complexes formed by disordered biomolecules, including not only more realistic membrane mimicking systems, such as bicelles or nanodiscs[1,2,5], but also to other systems such as lipid droplets[3,4] or membrane-less organelles[17].

## Results and discussion
### Backbone $^{15}$N spin relaxation times of peptides in micelles from NMR experiments

To experimentally characterize the dynamics of the selected six peptides in SDS micelle systems, we measured $T_1$, $T_2$, and hetNOE spin relaxation times of $^{15}$N atoms that were specifically labelled in the peptide backbone in positions shown in Fig. 1a. All the labelled residues were visible in heteronuclear single quantum coherence (HSQC) spectra with the exception of labelled N-terminal residues of hMff(TA) and yFis1(TA). HSQC spectra with the assignments and resulting spin relaxation times are shown in Figs. 1b, c, respectively.

For $T_1$ spin relaxation time, we observe an increasing trend in the order of eElaB(TA) < eYqjD(TA) < Magainin 2 < hMff(TA) < yFis1(TA) < GWALP, while the trend in $T_2$ is exactly opposite. Mitochondria-directed tail anchor proteins, eElaB(TA) and eYqjD(TA) derived from E. coli proteins have lower $T_1$ values and higher $T_2$ values than hMff(TA) and yFis1(TA) from humans and yeast. For peripheral Magainin 2 peptide, $T_1$ and $T_2$ times lay between values for tail anchors, while transmembrane GWALP has higher $T_1$ and lower $T_2$ values than any other peptide. HetNOE values lie between 0.4 and 1.0 for all the studied peptides, and systematic differences between peptides are not observed.

Spin relaxation times of $^{15}$N are mostly sensitive to dynamics of N-H bonds in ps to ns range in magnetic field-dependent manner[7]. However, the

interpretation of molecular dynamics from spin relaxation times is not straightforward, particularly for peptides in micelles where the detergent environment affects peptide dynamics in a non-trivial manner and standard models for protein dynamics may not be valid. Therefore, it is not clear from experimental data whether distinct $T_1$ and $T_2$ values arise from differences in overall rotation or conformational dynamics of peptides, and what is the role of detergents in this. In the following sections, we show how this experimental data can be interpreted using MD simulation models where detergents are explicitly included.

### Predicting spin relaxation times of micelles from physical interactions between atoms using MD simulations

To interpret the molecular dynamics of peptide-micelle complexes from the measured NMR spin relaxation times, we set out to perform MD simulations that reproduce the experimental spin relaxation times without any further fitting for both the micelle detergents and peptides. Such models provide comprehensive interpretation for the experimental data, which enables the separation of contributions of micelle dynamics from peptides' internal and overall dynamics to the experimentally measured spin relaxation times. To this end, we first aim to perform simulations that correctly capture micelle dynamics and reproduce experimental deuterium spin relaxation times from the literature[18].

Because Amber-based force fields with water models derived from TIP4P were previously successful in such tasks for partially disordered proteins[10], we first simulated an SDS micelle in water with parameters from AmberTools[19]. However, this micelle ended up in a gel-like phase with slow dynamics and deuterium spin relaxation times diverging from experimental data obtained from the literature[18] (Fig. 2). On the other hand, another popular protein force field with SDS parameters available, CHARMM36[20], is parameterized with the TIP3P[21] based water model, which suffers from low water viscosity and overly rapid dynamics, leading to incorrect spin relaxation time values that do not relate well to data obtained by experiments[11]. This is indeed observed also in our simulations in Fig. 2. Therefore, we proceeded to use CHARMM36 parameters with OPC water model[22] which has a viscosity value in good agreement with experiments[23]. This combination has been previously shown to give reasonable results for bilayers and monolayers[24,25]. Indeed, SDS micelles simulated with CHARMM36 parameters and the OPC water model remain in a fluid-like phase and predict deuterium spin relaxation times that are in good agreement with experiments (Fig. 2). Therefore, we set up also simulations of peptides in micellar environments using the CHARMM36 parameters with the OPC water model.

### MD simulations predicting spin relaxation times of peptide-micelle complexes

Besides the force field parameters, the number of SDS molecules per micelle has to be set manually in molecular dynamics simulations of peptide-micelle complexes, because simulations of spontaneous aggregation in large systems are not feasible at atomistic resolution. For preliminary screening, we simulated each peptide in a micelle with the sizes of 40, 45 and 50 SDS molecules for about 300 ns. In this quick scan, systems with 50 SDS molecules reproduced the experimental spin relaxation data quite well for all the peptides except GWALP. Due to the substantial computational cost, we performed a more systematic study on the micelle size dependence only for hMff(TA) which indicated the strongest size dependence in the initial screening. For this, we simulated hMff(TA) in micelles with 40, 45, 50, and 60 SDS molecules. Each system was simulated for at least 3 $\mu$s and repeated 3 times from different initial configurations. The results in Fig. 3a show systematic but weak dependence of $T_1$ on micelle sizes, while $T_2$ and hetNOE spin relaxation times from differently sized micelles are mostly within the error bars. Based on these results, we ran three independent simulations of each peptide except GWALP with 50 SDS molecules for at least 3 $\mu$s each. Spin relaxation times from these simulations were close to experimental values with the exception of eElaB(TA) for which $T_1$ was slightly overestimated

and $T_2$ slightly underestimated (Fig. 3). Therefore, eElaB(TA) simulations were repeated also in micelles with 40 SDS molecules, which indeed gave $T_1$ and $T_2$ values significantly closer to experiments, see Fig. 3d.

To understand the origin of large $T_1$ and small $T_2$ values in GWALP experiments, we screened the dependence on micelle size up to 80 SDS molecules per micelle for this peptide. Furthermore, we investigated whether dimerization could explain distinct spin relaxation times for GWALP by simulating two peptides in micelles with different numbers of SDS molecules. Results in Fig. 3 show that we can reproduce the experimental $T_1$ spin relaxation data either with one GWALP peptide in a micelle with 80 SDS molecules or with two GWALP peptides in a micelle with 70 SDS molecules. For systems with two GWALP peptides in an SDS micelle, we observe two different scenarios: peptides either strongly interact with each other, creating a dimer that rotates in the micelle as one entity, or the two peptides continue to rotate independently, see Supplementary Fig. 1a, b. We observe slight differences between these two scenarios in terms of $T_1$ spin relaxation times. While two independently rotating GWALP peptides in a micelle with 70 SDS molecules reproduce the experimental results slightly better, peptides rotating in a correlated manner would probably reproduce the experimental data equally well after a slight decrease in the number of SDS molecules. To also check the effect of dimerization on spin relaxation times in other peptides, we run simulations with two eElaB(TA) or yFis1(TA) molecules in one micelle. Differences between systems having one or two peptides in a micelle were smaller for eElaB(TA) and yFis1(TA) than for GWALP (Fig. 3b–d), and interactions between two peptides were observed neither for yFis1(TA) nor eElaB(TA), see Supplementary Fig. 1. In conclusion, our results suggest that distinct spin relaxation times for GWALP systems in experiments can be explained by their presence in larger aggregates, yet we cannot distinguish with the current data whether there are one or two peptides in each micelle. However, GWALP dimers seem slightly more probable than dimers of other peptides as GWALP peptides sometimes dimerize spontaneously and the dimers remain stable in simulations.

Spin relaxation times and representative snapshots from the systems that predict values closest to experiments are shown in Fig. 4a, b, respectively. These simulations reproduce the main experimentally observed differences in spin relaxation times between the peptides, particularly the increase of $T_1$ values in the order of eElaB(TA) ≲ eYqjD(TA) ≲ Magainin 2 < hMff(TA) ≲ yFis1(TA) < GWALP. Notably, after selecting the force field parameters and the number of molecules in the simulation system, no further fitting is made to reproduce the experimental data. Therefore, the selected simulations predict experimental spin relaxation times directly from physical interactions between atoms with relatively good accuracy, which justifies their further usage in interpreting the dynamic landscape of peptides in micellar environments performed in the next sections.

### Rotational dynamics of peptides in micelles

To characterize the timescales of proteins' rotational motions in micelles, we exploit the weights of different timescales resulting from the fit of exponential functions (Eq. (2)) to the rotational correlation functions of N-H (Eq. (1)) calculated from simulations. We consider the combination of weights and timescales from simulations that best reproduce experimental spin relaxation times in Fig. 4 as an interpretation of the dynamic landscape detected by the experiments. Notably, this direct approach (i) is free from the assumptions about the number of relaxation processes or their timescales, (ii) does not require the re-scaling of inaccurate simulation data that is required when using common methods to interpret dynamics from spin relaxation data[8,9,26–28], and (iii) provides a higher resolution and intuitively more comprehensible interpretation than the dynamical detector analysis[29]. We have previously reported similar analyses for pure protein[10,11] and lipid[30] systems.

The full dynamic landscapes for proteins with all observed timescales and their weights are shown in Supplementary Fig. 2, while a more comprehensible presentation where the weight of each timescale is represented

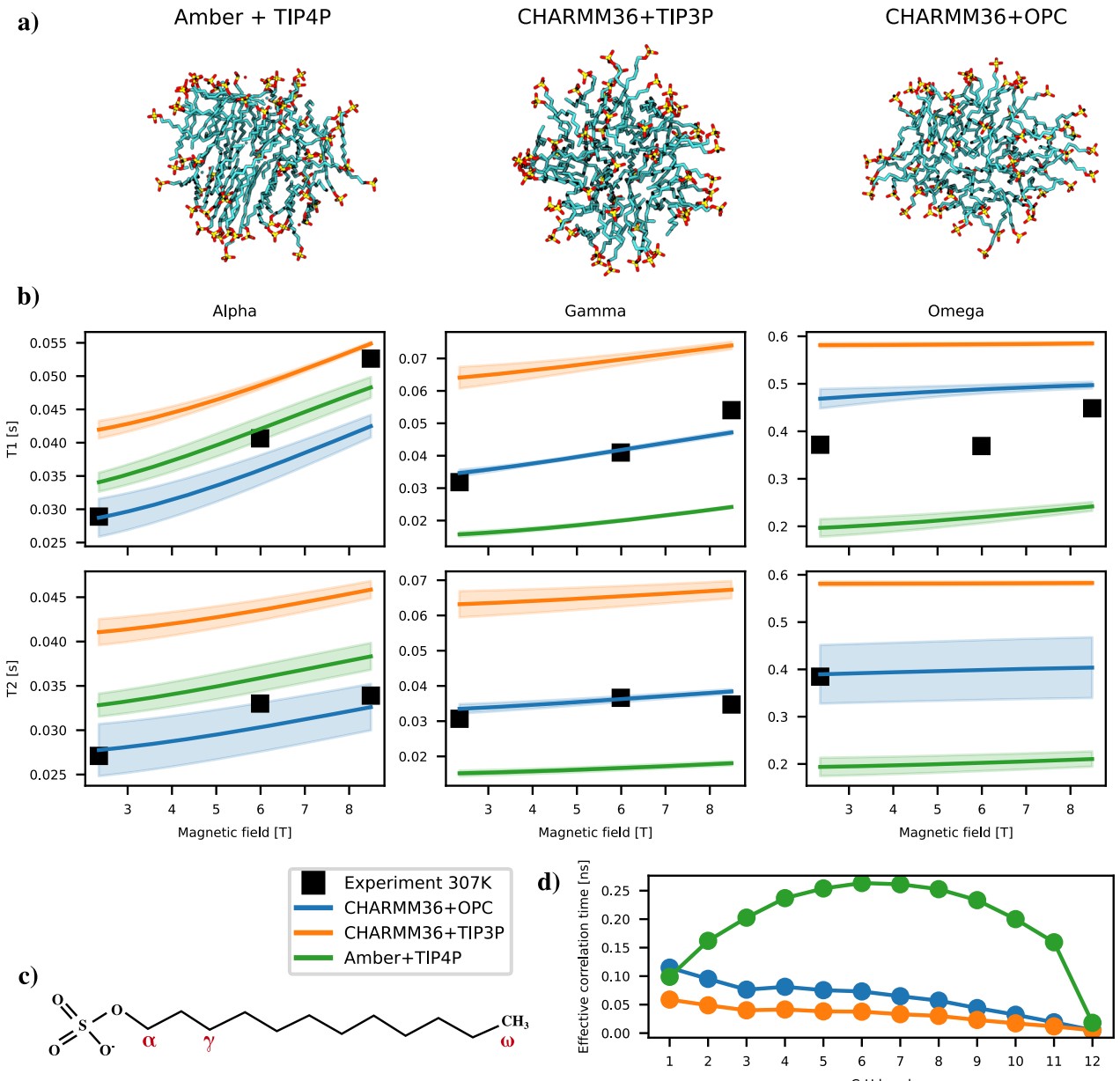

**Fig. 2 | Spin relaxation times of SDS micelles in water at 307 K. a** Snapshots from MD simulations showing the gel-like phase for Amber simulations and liquid-like phase for CHARMM36 simulations. **b** Deuterium $T_1$ and $T_2$ spin relaxation data from experiments[18] and MD simulations for isotopically labelled $\alpha$, $\gamma$ and $\omega$ segments. **c** Chemical structure of SDS with the assignment of labelled segments. **d** Effective correlation times, $\tau_{\text{eff}}$, of each C-H bond in SDS molecules from MD simulations.

by the point size is shown in Fig. 5a. For all the peptides, we observe dominant rotational timescales between approximately 5 and 9 ns, with weights above approximately 0.5 for most residues. Because these timescales are similarly present for almost all the residues within the same protein, we interpreted them to correspond to the overall rotational dynamics of peptides which can be described by one timescale in this case. Exception is the magainin 2 for which the unfolded region (residues 13–22) does not share the dominant timescales with others. Most residues in unfolded regions of tail anchor peptides (approximately residues 1–8) have the common dominant timescale with other residues, yet with slightly smaller weight. While we assign here a timescale for the overall protein motion, caution should be exercised when interpreting motions of peptides bearing significant disorder for which concepts developed for rigid body rotation are not valid. Because all proteins studied here (except GWALP) have large disordered fractions with respect to their size, we believe that further

analyses and interpretations of peptide dynamics are better to do using direct analyses from MD simulations (as exemplified in the following sections) than from components of rigid body rotation.

Nevertheless, we use the observed dominant timescales in Fig. 5a to interpret qualitative differences in overall rotations between different peptides. The fastest dominant timescales around 5 ns are observed for eElaB(TA) and eYqjD(TA), and Magainin 2, although the C-terminal half of Magainin 2 has even faster timescales around 4 ns. Dominant timescales of hMff(TA) and yFis1(TA) are slightly slower with values just above 6 ns, while GWALP exhibits significantly slower rotational dynamics than other peptides with dominant timescales of approximately 8 ns. Because the differences in dominant timescales between peptides correlate with the differences in $T_1$ and $T_2$ times, we conclude that the experimentally observed differences in $T_1$ and $T_2$ values arise mainly arise from differences in overall rotational dynamics between peptides.

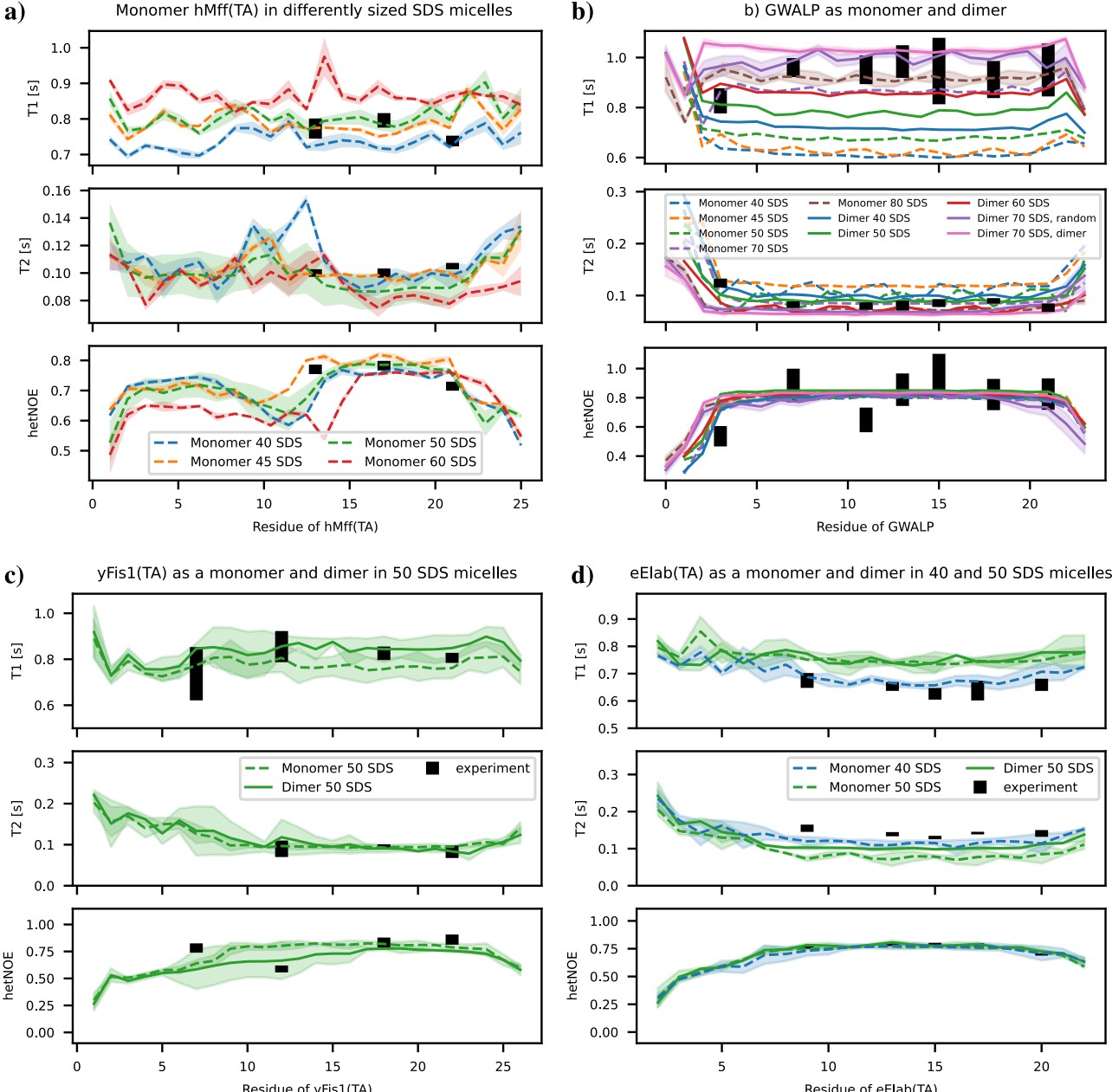

**Fig. 3 | Effect of micelle size and dimerization on spin relaxation times. a** Spin relaxation times of hMff(TA) peptide as a function of SDS micelle size. The dashed line shows the average over 3 replicas simulated with 1 peptide per micelle. The shaded region is the standard error of the mean calculated from the 3 simulations. **b** Spin relaxation times of GWALP peptide as a monomer or dimer as a function of the SDS micelle size. **c** Spin relaxation times of yFis1(TA) peptide in a micelle with 50 SDS molecules as a monomer or dimer. For the dimer, a solid line shows the average over the 2 peptides in a micelle in one simulation. For the monomer, the dashed line is the average taken over 3 replicas. Shaded regions are the standard errors of the mean. **d** Spin relaxation times of eElaB(TA) peptide in a micelle with 50 SDS molecules as a monomer or dimer, and in a micelle with 40 SDS molecules as a monomer.

## Peptide and surfactant rotations are uncoupled in micelles

While peptide rotations are dominated by timescales between approximately 5–9 ns corresponding their overall rotations, analysis of dynamic landscapes of SDS molecule C-H bonds reveals substantially different behaviour Fig. 5b: dynamics is dominated by timescales below 100 ps, and nanosecond timescale motions related to overall rotation of peptides are not observed for detergents. To further elucidate SDS rotation in micelles, distributions of SDS molecule overall rotation timescales calculated from vectors from micelle center of mass to sulfate atoms in SDS molecules are shown in Supplementary Fig. 3. Overall rotation timescales of individual SDS molecules exhibit wide distribution in all systems with most molecules having timescales faster than approximately 4 ns. Slower timescales with small weights appear only in systems with peptides. The wide distribution of timescales for individual SDS molecules and the lack of common dominant timescales with peptides suggest that their overall motions are not concerted. On the other hand, the appearance of slow timescales with small weights in systems with peptides indicate that few SDS molecules are attached to peptides such that they rotate partially together. Nevertheless, because the rotation of the clear majority of SDS molecules is not coupled with peptides, we conclude that peptides rotate independently from detergent molecules in a viscous media formed by the micelle.

Our results suggest that peptide and surfactant rotations do not share common timescales in micelles, and therefore a common timescale that would describe the rotation of a micelle as a whole cannot be defined.

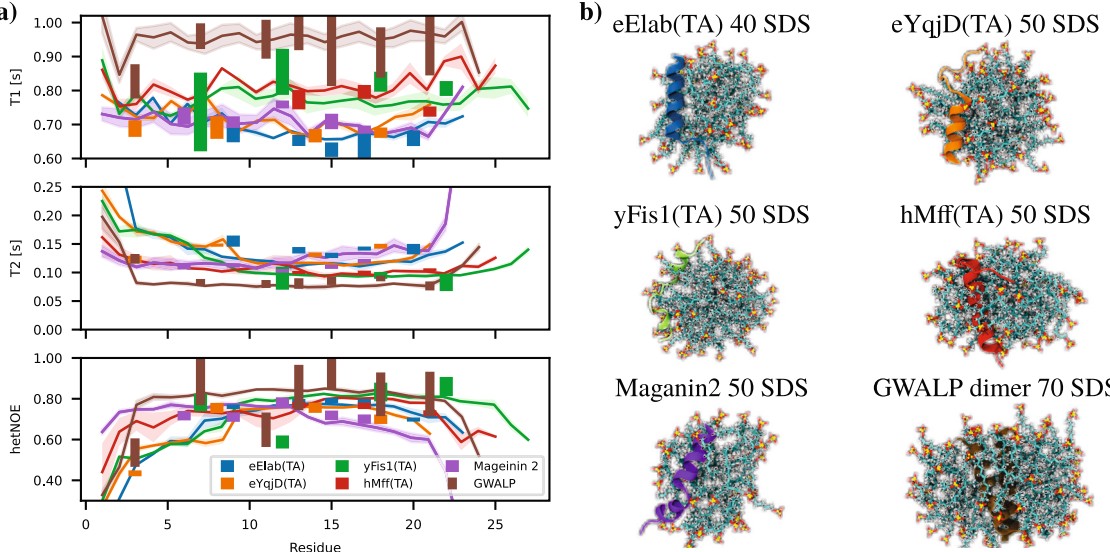

**Fig. 4 | Spin relaxation times from the best simulations compared with experiments. a** Spin relaxation times from the best simulations and experiments. Experimental values are in the middle of the shown rectangles and the edges represent the experimental error. The lines represent an average over 3 simulation replicas and the shaded region shows the error of the mean. **b** Representative snapshots of the studied peptides in SDS micelles.

Nevertheless, for peptides with rigid structures, it is possible to remove the peptide rotation from the trajectories and then analyze surfactant rotation. To further demonstrate uncoupling of peptide and surfactant rotations in micelles, we removed peptide rotation from trajectories of GWALP monomers in micelles with different amounts of SDS molecules, and then compared SDS overall rotation from these trajectories with the original trajectories. Results in Supplementary Fig. 4 show similar SDS rotations in trajectories with and without peptide rotation removal, providing further evidence that peptide and surfactant rotations are uncoupled in micelles.

We further investigated the coupling between micelle and peptide dynamics by comparing the peptide dynamics observed in MD simulations to the prediction from the Stokes-Einstein equation, assuming that the peptide-micelle complex rotates as a rigid body with a fixed radius in an environment having the viscosity of water. The Stokes–Einstein equation predicts significantly faster peptide dynamics and stronger dependence on the micelle radius than is observed in simulations, as shown in Fig. 6. For example, in the case of the hMff system in 50 SDS molecules, the gyromagnetic radius calculated from the simulations is 1.6 nm. However, to obtain the rotational timescales observed in simulations from the Stokes-Einstein equation, the radius of the micelle would have to be 3.0 nm, which is almost twice as large as the value from the simulations. On the other hand, the viscosity value in Stokes–Einstein equation should be 7.5 mPa s to obtain the same dynamics as observed in simulations, which is approximately ten times larger than the viscosity of water at 310 K (approximately 0.69 mPa s).

In conclusion, our results indicate a dynamic conception of peptide-micelle complexes where the rotational dynamics of peptides is dominated by timescales of a few nanoseconds related to their overall motion that can be experimentally detected by $T_1$ values. Because peptides rotate independently from detergents in a viscous media formed by the micelle, the rotation of the peptide-micelle complex cannot be described by the Stokes-Einstein equation that assumes that peptides and detergents rotate together as a spherical rigid body.

**Correlations between spin relaxation times and peptide secondary structure in micelles**

While the peptide overall rotation explains the experimentally observed differences in $T_1$ and $T_2$, spin relaxation times vary also along the sequence within a protein. Spin relaxation times measured here are sensitive to the rotational dynamics of peptide backbone N-H bonds, yet these dynamics depend indirectly also on conformations sampled by the peptides.

Therefore, differences in spin relaxation times along sequence are potential proxies also for conformational ensembles of peptides which is the case for example for partially disordered proteins[10]. To investigate if spin relaxation times could be useful to characterize peptide conformations also in the micellar environment, we analyzed correlations between peptide helicity and spin relaxation times in simulations (Fig. 7).

In simulations of individual systems, such as yFis1(TA) peptide in a micelle with 50 SDS molecules shown in Fig. 7a, helical regions have high hetNOE values and lower $T_2$ values than non-helical regions, while the changes in $T_1$ values are less clear. To investigate the generality of such correlations, we plotted the spin relaxation times as a function of the peptide helicity from all simulations listed in Supplementary Tables 1, 2 into Fig. 7b. This analysis reveals that helicity correlates with $T_2$ and hetNOE values with Pearson correlation coefficients of $-0.57$ and $0.79$, respectively, while the correlation with $T_1$ values is weaker with a Pearson correlation coefficient of $-0.15$. However, spin relaxation times depend also upon other properties besides helicity that vary between systems, such as micelle size. Therefore, we also calculated Pearson correlation coefficients between helicity and spin relaxation times separately for each individual simulation in Fig. 7c. In all individual simulations, correlation coefficients between helicity and het-NOE values are above 0.5, with $p$-values below 0.05. In the case of $T_2$ values, all the systems have correlation coefficients with helicity below $-0.45$ with $p$-values below 0.05 except for three systems (two replicas of hMff(TA) with 45 SDS molecules and a GWALP monomer with 50 SDS molecules) for which significant correlation was not found (correlation coefficients around $-0.25$ with $p$-values around 0.2). In the case of $T_1$ values, negative correlation with helicity with correlation coefficients below $-0.5$ and $p$-values below 0.05 are common, yet significant correlation is not found in many systems and some systems have also significant positive correlation. On the other hand, $T_1$ has a strong positive, $T_2$ mild negative, and netNOE very mild positive correlation with the micelle molecular weight in Supplementary Fig. 5, in line with conclusions in the above sections.

Observed correlations in simulations suggest that the residues with large hetNOE and small $T_2$ values have a higher tendency to form helices, particularly with respect to other residues within the same peptide, whereas correlations of $T_1$ values with the helical tendency is less straightforward. Such correlations motivate more detailed comparisons of spin relaxation time changes along the sequence between simulation and experiments. The central part of hMff(TA) and beginnings of eElab(TA), eYqjD(TA), and yFis1(TA) are unfolded in simulations and exhibit increased $T_2$ and

**a)**

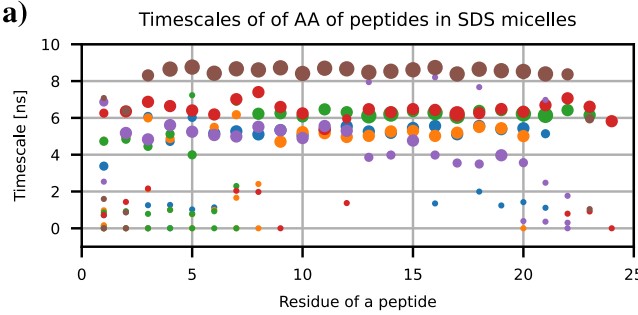

**b)**

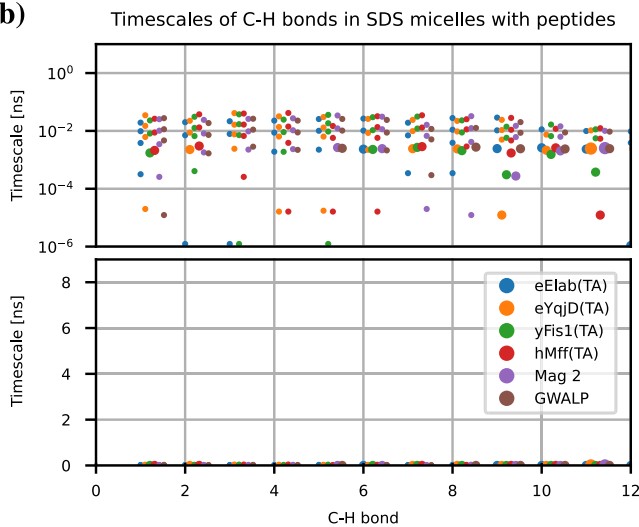

**Fig. 5 | Dynamics landscape of micelles with peptides.** Dynamic landscape of (**a**) peptides and (**b**) SDS molecules from the simulations in the best agreement with experiments in Fig. 4. The point sizes represent the weight of each timescale in the rotational relaxation process.

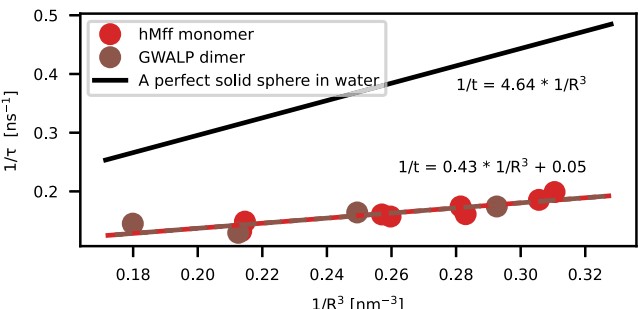

**Fig. 6 | Rotational dynamics of peptides as a function of micelle size.** Characteristic timescales from Stokes–Einstein equation (black line) and MD simulations for hMff(TA) monomer (red) and GWALP dimer (brown) in micelles with different numbers of SDS molecules. *X*-axis shows the radius of gyration.

decreased hetNOE values, but $^{15}$N labeled residues in experiments are too sparse to fully evaluate this observation. Magainin has reduced alpha-helical tendency at the end of the sequence associated with slight increase of $T_2$ and a decrease of hetNOE. These changes are not visible in experiments suggesting that simulations may not fully capture magainin secondary structure. Changes in spin relaxation times between residues are not observed in GWALP suggesting helical structure, with the exception of low hetNOE value of residue 11 in experiments. This may indicate some helical instability that is not visible in simulations, but could also be the experimental outlier.

In conclusion, our results suggest that inter-residual differences in spin relaxation times bear information on peptide secondary structure in a micelle which can be used to evaluate simulations and interpret experiments. However, accurate experimental data from multiple residues along the sequence are required for solid conclusion because relevant differences may be small.

## Conclusions

We show that MD simulations based on physical models can predict experimental peptide backbone $^{15}$N spin relaxation times in detergent systems with sufficient accuracy to interpret experiments without any further fitting. Our direct combination of MD simulation models and NMR data avoids indirect comparisons between two different models, i.e., a model used to interpret spin relaxation data vs. MD simulation model, which is done in many currently used methods[31]. On the other hand, our approach is free from assumptions about the number and timescales of rotation modes present in the system, as well as from arbitrary scaling of simulation results that are required to interpret spin relaxation time data and reproduce experimental results when deploying other methods[9,26–28,31]. These advances enable the interpretation of spin relaxation times for systems that are beyond

the scope of current approaches due to the large amounts of data required for parameter fitting, such as complex protein aggregates containing lipids or detergents.

To demonstrate the practical advances of our approach, we determined the dynamic landscape of peptide-detergent aggregates. Our findings support a view of peptide dynamics within a detergent matrix in which peptides and detergent molecules do not rotate together as a rigid body in a solvent. Rather, peptides rotate in a viscous medium composed of detergent micelle. Based on our interpretation of peptide backbone $^{15}$N spin relaxation times, the rotation of analyzed peptides in detergent aggregates was dominated by timescales between approximately 4–8 ns arising from the overall rotation of peptides. We explain the substantially slower overall rotation observed for transmembrane GWALP peptide, with the timescale of ~8 ns, by its preference for larger detergent aggregates than peripheral Magainin 2 or mitochondria tail anchor peptides having overall rotational timescales of ~4–6 ns. This result supports previous studies suggesting that mitochondria tail anchor peptides are more similar to peripheral peptides than transmembrane peptides[15,32–34]. On the other hand, the rotational dynamics of SDS molecules forming the detergents is dominated by timescales faster than 100 ps while ns timescales dominating in embedded peptides are absent.

Furthermore, our results elucidate indirect relations between peptide backbone $^{15}$N spin relaxation times, peptide dynamics and conformational ensembles. We found significant correlations of helical propensities of peptide residues with large hetNOE and low $T_2$ values, particularly when compared with the other residues within the same peptide. On the other hand, $T_1$ values mainly correlated with the overall rotation of peptides. These relations support rapid interpretations of peptide conformations in detergent aggregates from spin relaxation times even when MD simulation data is not available. On the other hand, these relations can be used to evaluate predictions from MD simulations against experiments.

The advantages of our direct combination of NMR experiments and MD simulations are demonstrated here for peptides in SDS detergent micelles, yet the presented approach can be applied to any biomolecular aggregate for which experimental spin relaxation times are accessible and realistic MD simulations can be performed. This includes many systems that are difficult to characterize by currently available experimental methods, such as fully or partially disordered proteins[10], bicelles or nanodiscs[1,2,5], membraneless organelles[17], and lipid droplets[3,4]. In addition to the interpretation of experimental NMR data, the presented approach will be useful also for the evaluation and improvement of MD simulation quality. In the era of data science and machine learning, such benchmark data is becoming increasingly important and endeavours to define such data for proteins and lipids are ongoing[35–37].

## Materials and methods
### NMR experiments
Peptides with the selected alanine, phenylalanine, glycine, or leucine having $^{15}$N labels in the backbone (positions shown in Fig. 1) were purchased from

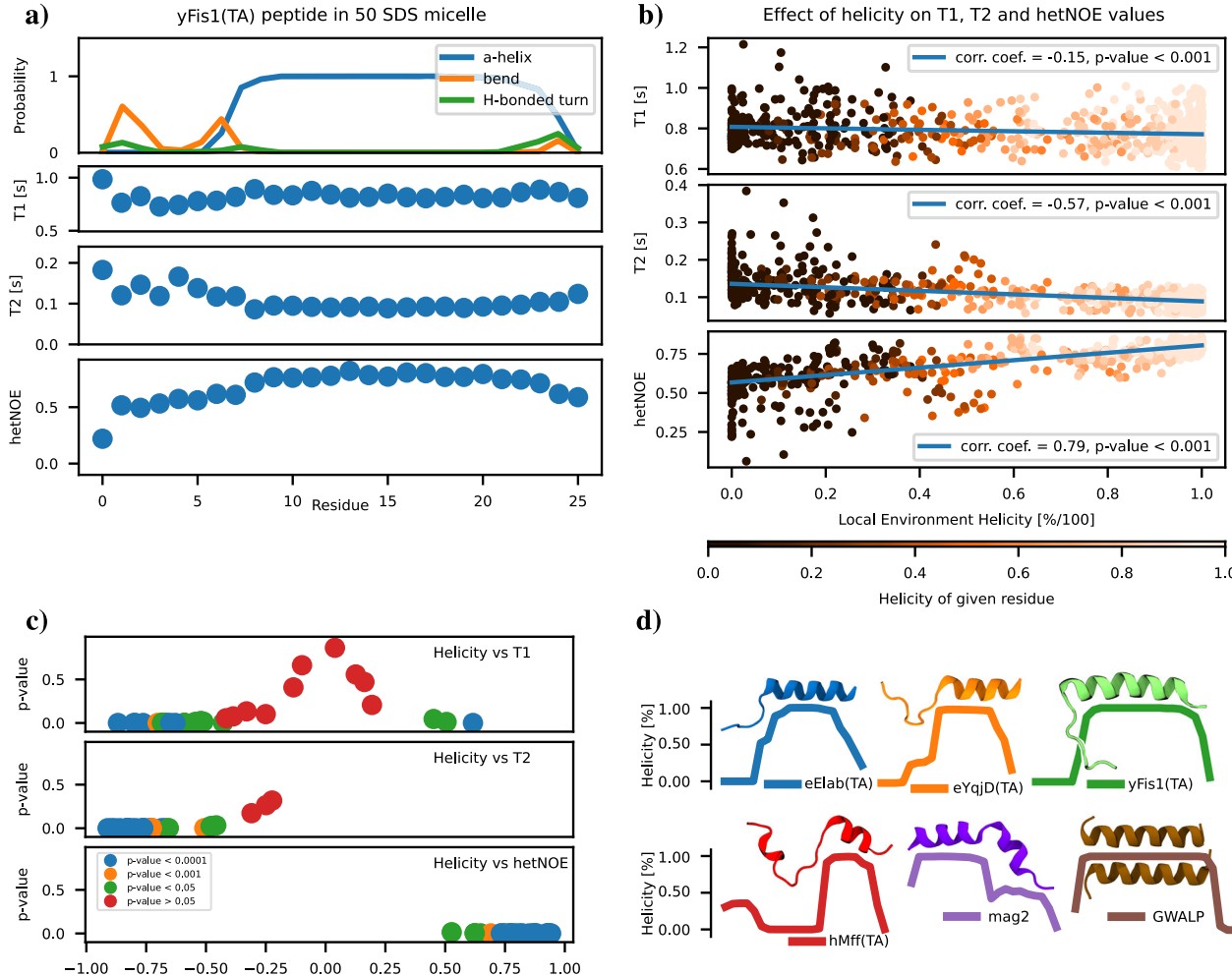

**Fig. 7 | Correlations between helicity and spin relaxation times of peptides.**
**a** Three most abundant secondary structures detected by DSSP[60] analysis and spin relaxation times from a simulation of yFis1(TA) in a micelle with 50 SDS molecules. **b** Scatter plot and Pearson correlation coefficients between helicity and spin relaxation times from individual residues in all simulations listed in Supplementary Tables 1, 2. The local environment helicity on the *x*-axis is the average over the given residue and the left and the right neighbouring residue if these exist. The colour of a point then encodes for the helicity of the given residue without neighbour averaging. **c** Pearson correlation coefficients and their *p*-values between the residual local environment helicity and spin relaxation times calculated separately for individual simulations. **d** Average helicities over three replicas with representative snapshots from MD simulations.

Peptide Protein Research Ltd (United Kingdom) in a powder form with the purity above 95%. Approximately 0.3 mg of each peptide was weighed in an eppendorf vial using an analytical balance (Precisa, XT 120A). The powder was then dissolved with deuterated sodium dodecyl sulfate (SDS, Sigma Aldrich) to obtain a solution with 0.3 mM peptide in 30 mM SDS and 20 mM sodium-phosphate buffer. 5% $D_2O$ was added for the lock in the NMR spectrometer. The solvent was prepared dissolving approximately 4.2 mg of SDS with 337.5 $\mu$l of milli-Q water, 90 $\mu$l sodium-phosphate buffer (pH 7.4, 0.1 M) and 22.5 $\mu$l of $D_2O$. The samples were transferred into 5 mm NMR tubes and all the NMR measurements were performed at 310 K using on Bruker Avance IIIHD 850 MHz spectrometer equipped with a cryogenically cooled probe head at the Institute of Biotechnology, University of Helsinki.

To assign the peaks from labelled amino acids, we measured [$^1$H,$^{15}$N]-HSQC (2048 points in the F3 domain, 128 points in the F2 domain, 16 scans, and recycling delay of 1.1 s between scans), [$^1$H,$^1$H]-TOCSY (1536 points in F3 domain, 512 points in F2 domain, 16 scans, mixing time of 60 ms), and [$^1$H,$^1$H]-NOESY spectra (1536 points in F3 domain, 512 points in F2 domain, 16 scans, mixing time of 280 ms). A recycling delay of 2.1 s was used between scans and the $^1$H carrier frequency was positioned at 4.703 ppm for both TOSCY and NOESY measurements. The spectral widths were 14.0 ppm ($^1$H, F1) and 10.2 ppm ($^1$H, F2) for [$^1$H,$^1$H]-TOCSY and 10.2 ppm ($^1$H,

F1) and 10.2 ppm ($^1$H, F2) [$^1$H,$^1$H]-NOESY. For HSQC spectra, the $^1$H carrier frequency was positioned at 4.703 ppm, the $^{15}$N carrier frequency at 119 ppm, and the spectral widths were 12.0 ppm ($^1$H, F2) and 33.0 ppm ($^{15}$N, F1). All these experiments were processed and analyzed using CcpNmr Analysis software (version 3.0.3)[38]. For the assignment, the complete spin systems of the amino acid residues were first identified using their proton-proton J-couplings ([$^1$H,$^1$H]-TOCSY) combined with their proton-nitrogen J-coupling ([$^{15}$N,$^1$H]-HSQC). These spin systems were then located within the peptide sequence by means of through-space, sequential NOE connectivities between adjacent residues[39].

Spin relaxation times were acquired with standard pulse sequences[7,40,41] using 1664 points in the F3 domain, 300 points in the F2 domain, and 8 scans. Delay times were set to 20, 50, 100, 200, 300, 500, 700, and 900 ms for $T_1$, and 34, 51, 68, 85, 119, 153, 187, 220, and 254 ms for $T_2$. The recycling delay of 3.5 s between scans was used for both $T_1$ and $T_2$, and 5 s for heteronuclear NOE. The spectral widths were the same as in HSQC experiments. The $T_1$ and $T_2$ relaxation data were processed and analyzed using Bruker Dynamic Center software (version 2.7.2). For the analysis of hetNOE spin relaxation times, peak heights were determined by TopSpin software from spectra with and without NOE. To determine the errors for hetNOEs, the signal-to-noise ratio was first determined from the region

without any peaks, and the limiting extremes of the noise values were then added to or subtracted from the peaks to estimate the largest effect of noise to the ratio between the peaks in the two spectra.

## MD simulations

**Simulated systems.** SDS micelles in water without peptides were simulated using Amber parameters from AmberTools[19] and CHARMM36[20] parameters from CHARMM-GUI[42,43]. Following the electronic continuum correction (ECC) to implicitly include electronic polarization, atom charges were scaled by factor 0.75 in Amber simulations[44] and compatible ion parameters were used[45]. Amber simulations were run with TIP4P[21] and CHARMM36 simulations with TIP3P[21] (CHARMM version), OPC[22], or TIP4P[21] water models. We first simulated SDS micelles at 298 K and 307 K with the standard saving frequency of 10 ps for coordinates for 362–1809 ns. Because SDS molecules had a substantial amount of rotational dynamics with faster timescales than the saving frequency, we initiated simulations with the saving frequency of 0.01 ps using conformations at different timepoints from the first simulations as the starting configurations. Simulated systems of SDS micelles without peptides and their starting configurations are listed in Supplementary Table 3.

To construct the initial configurations of peptides in micelles, peptide PDB files were first generated using ProBuilder server https://www.ddl.unimi.it/vegaol/probuilder.htm, and then embedded to SDS micelles using CHARMM GUI[42,43]. Systems were hydrated with approximately 24,000–40,000 water molecules and the total charge was neutralized with sodium ions. To test the potential dependence of the results on simulation box size, we ran hMff(TA) simulations with different box sizes, see Supplementary Fig. 6. We observed mild dependence on simulation box size with systems less than approximately 39,000 water molecules. This should be taken into account when planning the simulations, or concluding optimal micelle sizes from simulations with less amount of water than this. Simulation replicas were initiated from different time points of the first simulation for each system with a new random set of starting velocities. Further details are given in Supplementary Table 1.

Starting configurations for simulations with two GWALP, yFis1(TA), or eElaB(TA) peptides within the same micelle were prepared by (i) removing water molecules from an equilibrated snapshot of a monomeric system, (ii) adding a second peptide to close proximity of the micelle, (iii) solvating the system again. For the GWALP peptide, such simulations were run with 40, 45, 50, and 60 SDS molecules. In the simulation with 50 SDS molecules, two GWALP peptides started to interact with each other and created a dimer where the two peptides rotated together in the micelle, see Supplementary Fig. 1d). Therefore, with the optimal micelle size of 70 SDS, we ran replicas starting from conformations with (together) and without (separate) mutual interactions of two GWALP peptides. In all of the "together" simulations, peptides stayed in the form of the dimer for the whole length of the simulations. One of the systems with a "separate" starting configuration indicated potential dimer formation at the end of the simulation, while peptides retained independent motions otherwise. Simulations with systems having two peptides in the same micelle are summarized in Supplementary Table 2.

The convergence of systems was monitored by calculating the number of SDS molecules in each micelle as described below, for example of equilibrated system see Supplementary Fig. 7. We also ensured by visual inspection that all peptides were incorporated into micelles. Only the converged parts of trajectories were used for the analyses.

**Simulation details.** All simulations were performed using Gromacs versions 2021.1, 2021.5, and 2022.2[46,47]. Parameters from AmberTools were converted to Gromacs format using ACEPYPE[48]. Standard CHARMM-GUI equilibration procedure was used for all systems with the initial structure generated using CHARMM-GUI[42,43]. Replicas and other simulations initiated from already equilibrated configurations were started using randomly generated velocities without any further

equilibration. Energy for Amber simulations with initial structures from CHARMM simulations were minimized before starting the simulation whenever required.

For CHARMM36 simulations, timestep of 2 fs was used, the temperature was coupled using a Nosé-Hoover thermostat[49,50], the pressure was set to 1 bar with isotropic Parrinello-Rahman barostat[51], particle mesh Ewald (PME) was used for electrostatic interactions at distances longer than 1.2 nm[52,53], and Lennard-Jones interactions were cut off at 1.2 nm.

For simulations with Amber parameter, the timestep of 2 fs was used, the temperature was coupled using v-rescale thermostat, the pressure was set to 1 bar using isotropic Parrinello-Rahman barostat[51], PME was used to calculate electrostatic interactions at distances longer than 1.0 nm[52,53], and Lennard-Jones interactions were cut off at 1.0 nm.

**Calculation of spin relaxation times from MD simulations and interpretation of underlying timescales.** To couple spin relaxation times and molecular dynamics, we used Redfield equations[54] which connect $T_1$, $T_2$ and hetNOE spin relaxation times to the Fourier transformation (Spectral density) of the second-order rotational correlation functions of N-H, C-H or C-D bonds[11,18]. We calculated the rotational correlation functions, $C(t)$, for peptide backbone N-H bonds and C-H bonds in SDS molecules using the equation implemented in the gromacs package (gmx rotacf)[55]

$$C(t) = \left\langle \frac{3}{2}\cos^2\theta_{t'+t} - \frac{1}{2} \right\rangle_{t'},$$
(1)

where $\theta_{t'+t}$ is the angle between bond vector at the times $t$ and $t'$. To calculate the spectral density, we fitted a sum of exponential functions with the large number, $N$, of pre-fixed timescales, $\tau_i$, to the correlation functions from simulations using the Python scipy optimize.nnls solver:

$$C_{\text{fit}}(t) = \sum_{i=1}^{N} \alpha_i e^{-t/\tau_i}.$$
(2)

For peptide N-H bonds, $N = 100$ and $\tau_i$ values were equidistantly spaced in logarithmic scale between 1 ps and 100 ns. For SDS C-H bonds with a substantial amount of dynamics below 1 ps timescales, $N = 500$ and $\tau_i$ values were equidistantly spaced in logarithmic scale between 1 fs and 1 $\mu$s. As a result, the fitting gives the weight, $\alpha_i$, for each timescale that represents the relevance of the given timescale for the rotational relaxation of the bond. Spectral density, $J(\omega)$, is then obtained from the analytical Fourier transformation

$$J(\omega) = 2\int_0^\infty C_{\text{fit}}(t)cos(\omega t)\mathrm{d}t = 2\sum_{i=1}^{N} \alpha_i \frac{\tau_i}{1 + \omega^2\tau_i^2}$$
(3)

and substituted into Redfield equations[11,54]. The spin relaxation time calculation is implemented in the python code available at https://github.com/nencini/NMR_FF_tools/tree/master/relaxation_times.

Correlation functions up to lag times ($t$ in Eq. (1)) of one-hundredth of the total simulation length were used when analyzing the N-H peptide bonds, which should provide good statistics when analyzing single molecule simulations[56]. For C-H bonds in SDS molecules, averaging over a larger number of molecules enables usage of lag times up to one-twentieth of the total simulation length. Small but non-zero weights (below ~ 1% in all systems except micelle simulations with Amber in gel-like phase with the weights below ~ 10%) for the slowest possible timescale (100 ns for peptide N-H bonds, 1 $\mu$s for SDS C-H bonds) were observed for some correlation functions. These artificial timescales, arising from incomplete equilibration of correlation functions to plateau to zero, were not taken into account in the analyses, although ignoring them did not have major effects on the spin relaxation times.

To comprise dynamic landscapes of peptides and detergent molecules in micelles, we describe the relevance of different timescales for rotational

relaxation processes in different parts of molecules using the weights ($\alpha_i$) for timescales ($\tau_i$) resulting from the fit of Eq. (2) to the correlation functions calculated from simulations using Eq. (1). A similar analysis in our previous studies[10,11] gave dominant weights for the timescales reconciling with the overall rotation timescales for folded proteins, while most of the timescales had zero weights. For disordered protein regions, we observed more dispersed timescales without any dominant motion[10]. These results indicate that the approach can detect relevant dynamic processes without any further assumptions. Here we interpret dynamic landscapes of peptides and detergents in micelles using the weights of each timescale that result from the fitting to correlation functions from simulations that give the best agreement with experiments. To better emphasize the essential timescale ranges, we merged weights of five consecutive timescales for the plots of dynamic landscapes.

**Analysis of other properties**. Effective correlation times used to characterize the average dynamical timescales of SDS molecules in micelles were calculated as an integral over rotational correlation functions

$$\tau_{\text{eff}} = \int_0^\infty C(t)\mathrm{d}t \approx \sum_{i=1}^N \alpha_i \tau_i. \qquad (4)$$

To analyze the orientation of two peptides in the same micelle with respect to each other, we calculated the angle between the principal axes of the two peptides using the MDAnalysis package[57,58]. Rotational dynamics predicted by MD simulations were compared with the prediction for a spherical rigid body in a water media from the Stokes-Einstein equation

$$D_r = \frac{1}{6\pi\tau} = \frac{k_B T}{8\pi\eta r^3}, \qquad (5)$$

where $k_B$ is a Boltzmann constant, $T$ is the temperature, $\eta$ is the viscosity of water, $r$ is the radius, $D_r$ is the rotational diffusion coefficient, and $\tau$ is the timescale of the rotational dynamics of a rigid spherical object[59]. The radii of micelles were approximated using the radius of gyration. Because not all SDS molecules remain within a micelle throughout the simulation in some systems, we first determined which molecules are part of the micelle in simulations separately for each time step. This was done by selecting SDS molecules with any atom closer to any peptide atom than a cut-off distance of 1.4-1.8 nm. The cut-off distance was set system specifically to give the most reasonable results as exemplified for eYqjD(TA) simulation with 50 SDS molecules in Supplementary Fig. 7 where cut-off value 1.8 nm was used. The radii of gyrations were then calculated using these molecules together with the peptide(s) first separately for each configuration and then averaged over time. The number of SDS molecules in a micelle as a function of time was also used to monitor the equilibration of simulations as exemplified in Supplementary Fig. 7. The average numbers of SDS molecules within a micelle in each simulation are reported in Supplementary Tables 1, 2. These analyses were performed with Python scripts utilizing the MDAnalysis package[57,58].

Propensities of secondary structures in peptides were calculated using DSSP plug-in in Gromacs[55,60]. The propensity of individual secondary structural motifs was averaged over the time of trajectories. The local environment helicity propensity of a residue was correlated with the $T_1$, $T_2$ and hetNOE value of the residue for all peptides and simulations. Local environment helicity is defined as an average over a given residue and its left and right neighbour if these exist (the end residues have either only the left neighbour or only the right neighbour). The correlation was characterized by the Pearson correlation coefficient and the corresponding $p$-value using the Pearson function from the Python scipy.stats package.

## Data availability
Simulation data are available from references listed in Supplementary Tables 1, 2, 3. Data for figures are available from https://doi.org/10.5281/zenodo.10534985 and for correlation functions from https://doi.org/10.

5281/zenodo.8374967. All recorded NMR spectra are available from https://doi.org/10.5281/zenodo.10522640.

## Code availability
Codes to calculate spin relaxation times and create figures are available from https://doi.org/10.5281/zenodo.10534985.

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

## Acknowledgements
The facilities and expertise of the HiLIFE NMR unit at the University of Helsinki, a member of Instruct-ERIC Centre Finland, FINStruct, and Bio-center Finland are gratefully acknowledged. We acknowledge CSC – IT

Center for Science for computational resources. R.N., M.L.G.R., S.M.B., E.F. and O.H.S.O. thank the Academy of Finland for funding ((grant nos. 315596, 319902 and 345631)). C.D.D. acknowledges funding by the European Research Council (StG637649), the Academy of Finland (331556), the Jane and Aatos Erkko Foundation (200057) and the Sigrid Jusélius Foundation.

## Author contributions

R.N. performed and analyzed experiments and simulations, and wrote the manuscript together with OHSO. M.L.G.R. performed and analyzed MD simulations. S.M.B. performed and analyzed experiments and set up simulations. E.M. analyzed experiments. C.D.D. conceptulized the work together with O.H.S.O. O.H.S.O. conseptualized and superwised the work, and wrote the manuscript together with RN.

## Competing interests

The authors declare no competing interests.
