## [Peer Review File · Communications Chemistry]

Reviewers' comments:

Reviewer #1 (Remarks to the Author):

The manuscript "Probing the dynamic landscape of peptides in membrane mimics by synergized NMR experiments and MD simulations" describes the dynamics of micelle bound peptides by experiment and simulation. The text is written clearly and the concepts are presented in a very understandable manner. Also, the figures are presented clearly and the MD simulations are conducted rather well.

The author(s) report a detailed analysis of the peptide dynamics based on their MD simulations of the system. However, the analysis lacks a little bit in terms of generating more general models of the peptide motions and stick to analysis on a per amino acid level. It would have been interesting to see difference in general motion of the different types of peptides, like e.g., rotation about different peptide axes or bending motions.

Also, I like that the author(s) did compare their MD simulations to experimental data. However, here also lies my biggest problem with this paper because to me simulation and experiment do seem very disconnected. I do think that the experimental data unfortunately does not give much information about the peptide dynamics that the author(s) try to investigate. Looking at Fig. 1C it is obvious there is hardly any difference in the different labeled positions in any one peptide. Almost all the data are very flat lines. Making matters worse is the fact the even between peptides (that probably do have rather different dynamics) there are hardly any differences for T2 and hence. There are some differences in T1, but these are most likely due to different sizes of the micelles which the author(s) have to use to make the simulations fit the experimental data as shown in Fig. 3.

In the MD analysis the author(s) show that there should be some influence of peptide dynamics on relaxation rates, but they just don't show up in the experimental data. Just to give two particular examples:

1. Magainin 2 has two parts that are very different in structure (Fig. 7D) and dynamics (Fig. 5A) in the MD simulation but in Fig. 1C all lines of the experimental data for Magainin 2 are extremely flat.
2. Opposite to that GWALP does have some differences in the experimental data for different labeled positions (Fig. 1C) but in the MD analysis shows up as extremely flat over the sequence (Fig. 5A).

So, in summary to me it seems the experimental data is mostly indicative of micelle overall tumbling instead of any peptide dynamics. The trend is that for the peptides that do have higher

T1 at the same time T2 is low. The same trend can be observed for different micelle sizes in Fig. 3. It would have been beneficial to experimentally investigate micelle size for the different peptides (there is a factor of 2 in volume between them) with light scattering methods or something similar to show that their assumption about different micelle sizes is correct. This means while the simulations do represent the experimental data this does not mean that they correctly represent peptide dynamics as stated by the authors: "Because the resulting combination of weights and timescales reproduce experimental spin relaxation times, they can be considered as an interpretation of the dynamic landscape detected by the experiments." Possibly, this could be shown better by separating peptide dynamics from micelle rotation by removing micelle rotation from the MD trajectories during processing. This would allow quantifying the contribution of the individual motions to the overall relaxation rates.

In general solution NMR of fast rotating micelles might not be the best method for investigating peptide dynamics since the overall micelle rotation obscures the dynamics of the peptides. In addition, micelles are spherical objects that have any meaningful normal or director like a flat membrane. This makes analysis extremely complicated since how would one distinguish peptide rotation from micelle rotation?

In addition, the findings are very limited by the choice of SDS which unfortunately is not a good membrane mimic. As a detergent it is unfolding proteins and most likely heavily influencing the properties of the peptides studied here. Other more lipid like detergents would have been a better choice.

Therefore, I cannot recommend publication of this article in Nat.Comm.Chem.

Some more minor comments:

Quote from the introduction: "the lack of generally applicable straightforward and robust sample preparation protocols complicates many practical applications of solid-state NMR experiments."

This is simply not true. Sample preparation of solid-state NMR samples in particular of lipid membranes with peptides is very straightforward. The only difficulty that sometimes (rarely) appears is finding an organic solvent where both lipids and peptide dissolve. Everything else in the sample preparation is absolutely standard and simple. Even for large membrane proteins established protocols exist and work quite well on a large number of proteins.

Figs. 1C and 3D: Non-integer residue numbers should be avoided on the axis numbering.

Fig. 1C: Magainin is misspelled.

Fig. 6: The color for GWALP should be brown to stay with the color scheme used throughout the paper.

In the M&M section this appears: "NosÃSHoover thermostat" and 2x "ParrinelloâSRahman"

Reviewer #2 (Remarks to the Author):

This paper reports the analysis of peptide dynamics in a SDS environment using solution NMR and MD simulations.

This paper uses a thorough approach to characterise the complex dynamics of small peptides in SDS micelles. I concur with the authors that comprehensive MD simulations in combinations are an ideal means to analyse peptide dynamics in membranes.

Unfortunately, I am afraid that I am not convinced that SDS micelles are fair mimics of actual biological membranes. The reasoning of the authors appears somewhat paradox to me – they pursue a very accurate analysis of dynamics in a very artificial, unsuitable environment. I do not see how this will substantially improve our understanding of peptide dynamics in biological membranes.

I am afraid that I do not recommend publication of this manuscript, unless the authors could show at least for one of their systems that the behaviour in SDS is representative for membranes. This could be done by MD simulations or by solid-state NMR, and I strongly encourage the authors to provide such data (most likely MD simulations) in a major revision of their manuscript.

Reviewer #3 (Remarks to the Author):

The manuscript “Probing the dynamic landscape of peptides in membrane mimics by synergized NMR experiments and MD simulations” contains a comprehensive approach to combine MD and NMR relaxation methods to understand the dynamics of peptides in micellar environment. NMR relaxation data is used to calibrate and verify the MD setups and the MD data is used for a better interpretation of the NMR data. It is pleasing, that the authors made code available via github.

I have some concerns with the presentation and interpretation of the “dynamic landscape”. I’m wondering about the significance of the dynamic landscape presented for example in Figure 5 for claims like “peptides rotate independently from detergents”. More explicitly: The CH vectors of the SDS is very sensitive to local motion of the hydrocarbon sidechain and does not

necessarily reflect the motion of the SDS molecules in its entirety. Rotations around the long axis of SDS would for example modify the C-H vectors.

The N-H vector on the other hand is part of the ridged planar system of the peptide bond restricting its local motion considerably, especially within a secondary structure. Moreover, in a helix the N-H bond vector is quite close to the helix long axis. Rotations around the helix long axis is therefore not well reflected in the N-H correlation times. The integral/transmembrane peptides might rotate fast around their long axis without influencing the dynamic landscape defined by the N-H vector or the NMR relaxation.

However, in contrast to the NMR relaxation data, where all motion contributes to relaxation, the MD data (calibrated with the NMR data) could be used to distinguish between local motion, global motion and more. For example: It should be possible to extract the diffusion of the SDS headgroups on the surface of the micelles; transform this in a rotational correlation with the diameter of the micelles and compare with the rotation of the peptides around their long axis and perpendicular.

Reviewer #1 (Remarks to the Author):

The manuscript "Probing the dynamic landscape of peptides in membrane mimics by synergized NMR experiments and MD simulations" describes the dynamics of micelle bound peptides by experiment and simulation. The text is written clearly and the concepts are presented in a very understandable manner. Also, the figures are presented clearly and the MD simulations are conducted rather well.

AUTHOR REPLY:

We thank reviewer for positive and constructive feedback that have helped us to substantially improve our manuscript.

We have now substantially revised the manuscript to address the concerns raised by the reviewer. Particularly, we have clarified the main message of our manuscript and extended the discussion sections. The manuscript with changes marked is enclosed.

REVIEWER:

The author(s) report a detailed analysis of the peptide dynamics based on their MD simulations of the system. However, the analysis lacks a little bit in terms of generating more general models of the peptide motions and stick to analysis on a per amino acid level. It would have been interesting to see difference in general motion of the different types of peptides, like e.g., rotation about different peptide axes or bending motions.

AUTHOR REPLY:

In our previous works, we have analyzed also rotation about different peptide axes for folded protein regions, see Refs. 10-11. We performed similar analyses also for the peptides in micelles studied in this work. However, it turned out that peptides' secondary structures were too unstable to reasonably determine peptide axes for all peptides studied here except for GWALP. This is not surprising as significant fractions other proteins are unfolded in Fig. 7d. For this reason, we have opted originally to exclude these results from our manuscript. Nevertheless, we have now substantially extended discussion related to this. This discussion was previously part of "*Peptide and surfactant rotations are uncoupled in micelles*" section, but is now separated to its own section titled "*Rotational dynamics of peptides in micelles*".

REVIEWER:

Also, I like that the author(s) did compare their MD simulations to experimental data. However, here also lies my biggest problem with this paper because to me simulation and experiment do seem very disconnected. I do think that the experimental data unfortunately does not give much information about the peptide dynamics that the author(s) try to investigate.

AUTHOR REPLY:

In the manuscript, we directly calculate the spin relaxation times from MD simulations without any additional fitting of the parameters that describe the physics of systems. We show that it is possible to reproduce experimentally observed differences in spin relaxation times between peptides by setting correct number of molecules in simulations. We then use simulations to interpret the experiments, which is not possible with other available methods. Based on this analysis, we conclude, for example, that experimentally observed differences in T_1 and T_2 between peptides result from differences in overall dynamics of the peptides, and SDS and peptide molecule rotations are uncoupled. These conclusions are possible because we directly combine MD simulations and NMR data for both peptides and SDS molecules. NMR data is used to validate the simulations and MD simulation data is used to interpret the experiments. Therefore, we think that simulations are

very much connected with the experiments in the manuscript, and that this combination delivers significant and novel information of peptide dynamics in micellar environment.

REVIEWER:

Looking at Fig. 1C it is obvious there is hardly any difference in the different labeled positions in any one peptide. Almost all the data are very flat lines. Making matters worse is the fact the even between peptides (that probably do have rather different dynamics) there are hardly any differences for T₂ and hence. There are some differences in T₁, but these are most likely due to different sizes of the micelles which the author(s) have to use to make the simulations fit the experimental data as shown in Fig. 3.

AUTHOR REPLY:

Using MD simulations to interpret the experiments, we conclude that experimentally observed differences in T₁ and T₂ times between peptides can be explained by differences in overall rotation dynamics of peptides. This is now clarified in section “*Rotational dynamics of peptides in micelles*” in the revised version. It is important to note that this is not a trivial conclusion, as now clarified also in the section “*Backbone 15N spin relaxation times of peptides in micelles from NMR experiments*”:

“... the interpretation of molecular dynamics from spin relaxation times is not straightforward, particularly for peptides in micelles where the detergent environment affects peptide dynamics in a non-trivial manner and standard models for protein dynamics may not be valid. Therefore, it is not clear from experimental data whether distinct T₁ and T₂ values arise from differences in overall rotation or conformational dynamics of peptides, and what is the role of detergents in this. In the following sections, we show how this experimental data can be interpreted using MD simulation models where detergents are explicitly included.”

It is true that major differences in spin relaxation times between labeled positions within a same peptide are not observed in figure 1C. Based on the results in section titled “*Correlations between spin relaxation times and peptide secondary structure in micelles*” in the revised manuscript, this would suggest that there are no major changes in the secondary structure along the sequence at the region where residues are labeled (for further discussion, see the reply to the next comment).

REVIEWER:

In the MD analysis the author(s) show that there should be some influence of peptide dynamics on relaxation rates, but they just don't show up in the experimental data. Just to give two particular examples:

1. Magainin 2 has two parts that are very different in structure (Fig. 7D) and dynamics (Fig. 5A) in the MD simulation but in Fig. 1C all lines of the experimental data for Magainin 2 are extremely flat.

2. Opposite to that GWALP does have some differences in the experimental data for different labeled positions (Fig. 1C) but in the MD analysis shows up as extremely flat over the sequence (Fig. 5A).

AUTHOR REPLY:

Analyses of simulations in section “*Correlations between spin relaxation times and peptide secondary structure in micelles*” reveal weak but statistically relevant correlations of alpha-helical tendency with T₂ and hetNOE values. In simulations we also observe significant instabilities of alpha-helices in some peptides that are associated with corresponding changes in the spin relaxation times. However, as pointed out by the reviewer, these changes are not visible in the experimental data. This could mean that there are small discrepancies in secondary structures between

simulations and experiments. On the other hand, labeled residues are quite sparse in our experiments and differences in spin relaxation times between residues in a same peptide are quite small (typically smaller than differences between peptides). Therefore, it is difficult to make very strong conclusions on this based on the current data. We originally included this discussion into the manuscript because our results indicate that evaluation and interpretation also of secondary structures will be possible in the future. For example, simulations with different initial structures or simulation parameters could be generated and compared with experiments having more residues labeled. This discussion is now extended and clarified in the section titled “*Correlations between spin relaxation times and peptide secondary structure in micelles*”.

REVIEWER:

So, in summary to me it seems the experimental data is mostly indicative of micelle overall tumbling instead of any peptide dynamics. The trend is that for the peptides that do have higher T₁ at the same time T₂ is low. The same trend can be observed for different micelle sizes in Fig. 3.

AUTHOR REPLY:

Based on our analyses, we conclude that experimentally observed differences in T₁ and T₂ times between peptides can be explained by differences in overall rotation dynamics of peptides. This is summarized in section “*Rotational dynamics of peptides in micelles*”:

“Because the differences in dominant timescales between peptides correlate with the differences in T₁ and T₂ times, we conclude that the experimentally observed differences in T₁ and T₂ values arise from differences in overall rotational dynamics between peptides”.

On the other hand, several analyses discussed in section “*Peptide and surfactant rotations are uncoupled in micelles*” indicate that peptide and surfactant molecules do not rotate together as a rigid body, but peptide rotates in a viscous environment formed by the micelle (see also replies to other reviewers’ comments). In other words, we conclude that T₁ and T₂ times are mostly indicative of the peptide overall rotation, but this is not the same as micelle overall rotation.

Importantly, these conclusions are not trivial and could not be convincingly made without directly combining MD simulations and experimental spin relaxation times. Also this is now clarified in the manuscript.

REVIEWER:

It would have been beneficial to experimentally investigate micelle size for the different peptides (there is a factor of 2 in volume between them) with light scattering methods or something similar to show that their assumption about different micelle sizes is correct.

AUTHOR REPLY:

It is important to note that differences in micelle sizes between different peptides is not an assumption in our work. Micelle size is effectively a parameter in the MD simulation model because number of SDS molecules per micelle has to be manually set. Here we optimize this parameter against experimental data. This is the main explanation for the differences of GWALP with respect to other peptides, but it does not fully explain differences between other peptides. Nonetheless, it would be indeed interesting to test this result with independent experiment. However, we are not confident that we would be able to measure sufficiently accurate light scattering data from particles with the radius of approximately 3 nm, which is the case for smallest micelles. X-ray scattering experiments might be better option, but they would require substantial additional effort. On the other hand, our main conclusion is that differences in peptide overall rotations (not necessarily the size) explain the differences in T₁ and T₂. Therefore, we have opted to leave the potential X-ray scattering experiments to the further studies.

REVIEWER:

This means while the simulations do represent the experimental data this does not mean that they correctly represent peptide dynamics as stated by the authors: "Because the resulting combination of weights and timescales reproduce experimental spin relaxation times, they can be considered as an interpretation of the dynamic landscape detected by the experiments." Possibly, this could be shown better by separating peptide dynamics from micelle rotation by removing micelle rotation from the MD trajectories during processing. This would allow quantifying the contribution of the individual motions to the overall relaxation rates.

AUTHOR REPLY:

It is important to note that ^{15}N spin relaxation times are measured from peptide backbone in the manuscript. Therefore, these values directly represent the dynamics of peptides. In the manuscript, we first show that we can reproduce the experimental spin relaxation times with MD simulations. Then we use these simulations to interpret which kind of peptide motions produce the experimentally measured spin relaxation times. Because our simulations predict experimental spin relaxation times of peptides in micelles from physical interactions between atoms, we consider the motions observed in simulations to be reasonable interpretation for how peptides rotate.

On the other hand, in section "*Peptide and surfactant rotations are uncoupled in micelles*", we demonstrate by three different analyses that the rotations of micelle molecules are not coupled with the peptide rotations. Therefore peptides do not rotate together with micelle molecules, but rather rotate in fluid like viscous environment formed by the micelle. Because overall rotation of such fluid-like micelle is not well defined, its contribution to peptide rotation cannot be analyzed.

REVIEWER:

In general solution NMR of fast rotating micelles might not be the best method for investigating peptide dynamics since the overall micelle rotation obscures the dynamics of the peptides. In addition, micelles are spherical objects that have any meaningful normal or director like a flat membrane. This makes analysis extremely complicated since how would one distinguish peptide rotation from micelle rotation?

AUTHOR REPLY:

If the aim is to understand peptide behavior in membranes, we agree with the reviewer that fast rotating micelles are not ideal systems. However, in practice, such systems are widely used to study membrane interacting peptides and proteins because preparation and experimentation of better membrane mimics is often not feasible in practice. Micellar samples for solution state NMR are often easier to prepare and measure, and require less amounts of sample than more membrane like systems for solid state NMR. Due to the wide use of micellar systems as model systems, it is highly useful to understand peptide dynamics also in such systems. More detailed understanding of model systems helps estimating the relevance of results for systems under primary interest.

More importantly, micelles as membrane mimics is only one class of fast rotating molecular aggregates that can be studied with the approach presented here. The approach can be applied, for example, to peptide micelles, bicelles, nanodiscs and lipid droplets that are difficult to characterize with other methods. SDS micelle-peptide systems is used here as an example system to demonstrate that we can characterize dynamics of small and fast rotating objects by combining MD simulations and solution state NMR experiments. Our currently ongoing studies for bicelles and nanodiscs will be published separately.

Thanks to reviewers' feedback we realized that these aspects were not clearly presented in the first version of the manuscript, where membrane mimics were too much in the focus, particularly in the title, abstract and introduction. We have now substantially revised the manuscript to better convey its main message. The title is now updated to "*Probing the dynamic landscape of peptides in molecular assemblies by synergized NMR experiments and MD simulations*". Abstract and introduction are revised to emphasize biomolecular aggregates more generally, whereas membrane mimics are mentioned as one example.

We agree with the reviewer that peptide dynamics in fast rotating micelles or other small molecular assemblies is indeed difficult to analyze. However, in our manuscript we demonstrate how this can be done by directly combining NMR experiments with MD simulations. We believe that this approach will be useful to understand wide range of systems from membrane mimics to nanodiscs and micelles with potential therapeutic applications.

REVIEWER:

In addition, the findings are very limited by the choice of SDS which unfortunately is not a good membrane mimic. As a detergent it is unfolding proteins and most likely heavily influencing the properties of the peptides studied here. Other more lipid like detergents would have been a better choice.

AUTHOR REPLY:

As mentioned also in the reply to the above comment, our primary goal was not to study membrane mimics to understand how peptides behave in membranes. Our main message is that the presented combination of MD simulations and solution state NMR experiments can be used to characterize dynamics of small and fast rotating molecular aggregates. SDS micelles are widely used membrane mimics and serve here as a test system for the developed methodology, but are just one example of the possible systems for which the developed methodology can be applied. In addition to membrane mimicking micelles, these include, for example, peptide micelles, bicelles, nanodiscs and lipid droplets that are difficult to characterize with other methods. Therefore, the quality of SDS micelles as membrane mimics is not critically relevant for the main messages of our manuscript. Furthermore, our conclusions related to the sensitivity of spin relaxation times on different peptide properties (T_1 senses mainly overall rotation, whereas T_2 and hetNOE correlate weakly with secondary structure) and uncoupling of peptide and surfactant motions are most likely valid also for other than SDS micelles.

We have now substantially revised the manuscript to better convey our main message as detailed in the reply to the previous comment.

REVIEWER:

Therefore, I cannot recommend publication of this article in Nat.Comm.Chem.

AUTHOR REPLY:

We thank reviewer again for the constructive comments. We believe that we have now substantially improved our manuscript, and have been able to answer all the concerns raised by the reviewer. We hope that it is possible to reevaluate the revised version for the Communications Chemistry.

REVIEWER:

Some more minor comments:

Quote from the introduction: "the lack of generally applicable straightforward and robust sample preparation protocols complicates many practical applications of solid-state NMR experiments."

This is simply not true. Sample preparation of solid-state NMR samples in particular of lipid membranes with peptides is very straightforward. The only difficulty that sometimes (rarely) appears is finding an organic solvent where both lipids and peptide dissolve. Everything else in the sample preparation is absolutely standard and simple. Even for large membrane proteins established protocols exist and work quite well on a large number of proteins.

AUTHOR REPLY:

We have now substantially revised the introduction to better convey the main message of our manuscript, and this sentence and point are now removed from the manuscript. However, we still think that in the cases where it is not clear how peptides interact with membranes (such as tail anchors in our work), it is very difficult to know which kind of sample preparation will work in practice, and optimizing this for solid-state NMR experiments requires a lot of work and material.

REVIEWER:

Figs. 1C and 3D: Non-integer residue numbers should be avoided on the axis numbering.

AUTHOR REPLY:

These are now changed to integers.

REVIEWER:

Fig. 1C: Magainin is misspelled.

AUTHOR REPLY:

This is now corrected.

REVIEWER:

Fig. 6: The color for GWALP should be brown to stay with the color scheme used throughout the paper.

AUTHOR REPLY:

Color of GWALP is now changed to brown also in Fig. 6.

REVIEWER:

In the M&M section this appears: "NosÃSHoover thermostat" and 2x "ParrinelloÃSRahman"

AUTHOR REPLY:

This is now corrected.

Reviewer #2 (Remarks to the Author):

This paper reports the analysis of peptide dynamics in a SDS environment using solution NMR and MD simulations.

This paper uses a thorough approach to characterise the complex dynamics of small peptides in SDS micelles. I concur with the authors that comprehensive MD simulations in combinations are an ideal means to analyse peptide dynamics in membranes.

AUTHOR REPLY:

We thank reviewer for positive and constructive feedback that have helped us to improve our manuscript. Based on the comments, we have now substantially clarified the main message of our manuscript.

REVIEWER:

Unfortunately, I am afraid that I am not convinced that SDS micelles are fair mimics of actual biological membranes. The reasoning of the authors appears somewhat paradox to me – they pursue a very accurate analysis of dynamics in a very artificial, unsuitable environment. I do not see how this will substantially improve our understanding of peptide dynamics in biological membranes.

I am afraid that I do not recommend publication of this manuscript, unless the authors could show at least for one of their systems that the behaviour in SDS is representative for membranes. This could be done by MD simulations or by solid-state NMR, and I strongly encourage the authors to provide such data (most likely MD simulations) in a major revision of their manuscript.

AUTHOR REPLY:

As mentioned also in the reply to the similar comment by reviewer 1, our primary goal was not to study membrane mimics to understand how peptides behave in membranes. Our main message is that the presented combination of MD simulations and solution state NMR experiments can be used to characterize dynamics of small and fast rotating molecular aggregates. SDS micelles are widely used membrane mimics and serve here as test systems for the developed methodology, but are just one example of the possible systems for which the developed methodology can be applied. In addition to membrane mimicking micelles, such systems include, for example, peptide micelles, bicelles, nanodiscs and lipid droplets that are difficult to characterize with other methods. Therefore, the quality of SDS micelles as membrane mimics is not critically relevant for the main messages of our manuscript, and comparisons between peptide behavior in SDS and membranes is beyond the scope of this work. We are currently applying the presented approach to bicelles and nanodiscs, but these studies will be published separately.

We have now substantially revised the manuscript to better convey its main message. The title is now updated to *“Probing the dynamic landscape of peptides in molecular assemblies by synergized NMR experiments and MD simulations”*. Abstract and introduction are revised to emphasize biomolecular aggregates more generally, whereas membrane mimics are mentioned as one example.

Reviewer #3 (Remarks to the Author):

The manuscript “Probing the dynamic landscape of peptides in membrane mimics by synergized NMR experiments and MD simulations” contains a comprehensive approach to combine MD and NMR relaxation methods to understand the dynamics of peptides in micellar environment. NMR relaxation data is used to calibrate and verify the MD setups and the MD data is used for a better interpretation of the NMR data. It is pleasing, that the authors made code available via github.

AUTHOR REPLY:

We thank reviewer for positive and constructive feedback that have helped us to substantially improve our manuscript. We have now revised manuscript based on reviewer’s comments and performed analyses suggested by the reviewer.

REVIEWER:

I have some concerns with the presentation and interpretation of the “dynamic landscape”. I’m wondering about the significance of the dynamic landscape presented for example in Figure 5 for claims like “peptides rotate independently from detergents”. More explicitly: The CH vectors of the SDS is very sensitive to local motion of the hydrocarbon sidechain and does not necessarily reflect the motion of the SDS molecules in its entity. Rotations around the long axis of SDS would for example modify the C-H vectors.

AUTHOR REPLY:

Original rationales for our argument that peptides rotate independently from detergents were that (I) common timescales related to the overall rotation of peptides that dominated N-H bond rotation were not present in SDS molecule C-H bonds, and (II) peptide rotation could not be predicted by the Stokes-Einstein equation. If a peptide and SDS molecules in a micelle would rotate together as a rigid object, they would share the same common dynamic timescales and the protein rotation should follow Stokes-Einstein equation.

Reviewer is suspecting that the lack of common timescales (argument I) follows from local motions of C-H bonds rather than the distinct overall rotation modes between SDS molecules and peptides. To ensure that this is not the case in our simulations, we calculated the correlation functions and timescales for the vectors between micelle’s center of the mass and SDS sulfur atom (similarly to the suggestion by the reviewer in another comment below). The distribution of timescales for this vector from different SDS molecules are shown now in figure S3 in the supplementary information. Also the discussion in “*Peptide and surfactant rotations are uncoupled in micelles*” section is largely revised and now reads:

“... Overall rotation timescales of individual SDS molecules exhibit wide distribution in all systems with most molecules having timescales faster than approximately 4 ns. Slower timescales with small weights appear only in systems with peptides. Wide distribution of timescales for individual SDS molecules and the lack of common dominant timescales with peptides suggests that their overall motions are not concerted. On the other hand, appearance of slow timescales with small weights in systems with peptides indicate that few SDS molecules are attached to peptides such that they rotate partially together. Nevertheless, because the rotation of clear majority of SDS molecules are not coupled with peptides, we conclude that peptides rotate independently from detergent molecules in a viscous media formed by the micelle.”

In conclusion, we have now added the third analysis in the manuscript that is further supporting the idea that peptides rotate independently from detergent molecules in micelles.

REVIEWER:

The N-H vector on the other hand is part of the ridged planar system of the peptide bond restricting its local motion considerably, especially within a secondary structure. Moreover, in a helix the N-H bond vector is quite close to the helix long axis. Rotations around the helix long axis is therefore not well reflected in the N-H correlation times. The integral/transmembrane peptides might rotate fast around their long axis without influencing the dynamic landscape defined by the N-H vector or the NMR relaxation.

AUTHOR REPLY:

Rotational correlation functions have quite complicated form with multiple timescales even in a rigid body with asymmetric shape, see for example Eq. (35) in <https://doi.org/10.1063/1.1701390>. Therefore, it would not be straightforward to assign specific timescales related to different component even for the rotation of rigid alpha-helix. We have previously characterized rotation of folded proteins using rigid body diffusion coefficients that can be more easily assigned to different motions (see Refs. 9-10). We tried similar analysis for peptides in this work, but it was not possible to reasonably assign regions that could be approximated as rigid bodies because peptides analyzed here are quite small and bear significant fractions of disordered regions. Therefore, we have opted not to include this analysis in the manuscript.

It is true that N-H bond rotations would be probably less sensitive to rotations around the long axis in systems with rigid alpha-helix. However, GWALP is the only peptide with clear rigid alpha-helices analyzed in this work. Therefore, we do not discuss on rigid body dynamics in the manuscript and this issue should not be relevant for our conclusions.

REVIEWER:

However, in contrast to the NMR relaxation data, where all motion contributes to relaxation, the MD data (calibrated with the NMR data) could be used to distinguish between local motion, global motion and more. For example: It should be possible to extract the diffusion of the SDS headgroups on the surface of the micelles; transform this in a rotational correlation with the diameter of the micelles and compare with the rotation of the peptides around their long axis and perpendicular.

AUTHOR REPLY:

As mentioned also in the replies above, the separation of contributing motions is not straightforward for molecules with significant disorder that cannot be approximated in terms of rigid body rotation. Importantly, we demonstrated in the manuscript that MD simulations can be particularly useful in such situations because they can deliver interpretations for dynamics independently from the rigid body rotation terminology. We have now elucidated this aspect in section “*Rotational dynamics of peptides in micelles*”.

Nevertheless, we have utilized the idea by the reviewer to compare rotation of SDS headgroups on the micelle surface with the peptide rotation to give further support for our result that peptide and SDS rotations are uncoupled, see the answer above, section “*Peptide and surfactant rotations are uncoupled in micelles*”, and Fig. S3 in the supplementary information. Conclusion is the same that previously: SDS molecules and protein do not share the same dominant timescales, thus their rotations are uncoupled.

Reviewers' comments:

Reviewer #1 (Remarks to the Author):

The manuscript "Probing the dynamic landscape of peptides in molecular assemblies by synergized NMR experiments and MD simulations" has been revised to address previous criticism. The authors acknowledged that their systems are not very good membrane mimics and changed the text and title accordingly. I think that this is a good decision since it emphasizes the more technical nature of this work.

However, my main criticism was not really addressed in my opinion. I thought that the experimental results are mostly representative of micelle rotation instead of peptide dynamics and the authors have not convinced me otherwise. They did not conduct any other experiments, simulations or even analysis which would have been rather easy to prove either the authors or my view. In Fig. 5 the authors have timescale data for each peptide but always for only one micelle size. It should have been easy to take the existing simulations for hMff(TA) with 40, 45, 50, and 60 SDS molecules and conduct the same analysis as in Fig. 5 for all four micelle sizes. If peptide and micelle rotation are independent the authors should get the same results for all of them since the peptide remains the same in all simulations and only the micelle size changes. I doubt that this would be the case though since Fig. 3a already shows that they lead to different predictions of experimental results. So judging from Figs. 1c, 3a and 3b, I still believe the almost all differences they see, originate from differences in micelle rotation.

The authors try to negate this interpretation by showing that peptide and micelle rotations are uncoupled. However, I think in particular the first analysis about the correlation times is very misleading and doesn't tell us anything about the question. Of course peptide and surfactant rotations are uncoupled. This is absolutely expected. The molecular weight of SDS is that of about 3 alanines. So all peptides are much heavier than a SDS molecule. Also internal flexibility of SDS is much higher than that of alpha-helices. One would have to compare peptide motion to micelle motion as a whole and not to SDS motion.

Imagine a membrane protein inside a membrane. Measuring any dynamic property of the lipids and the protein and comparing them, one will find the lipids are way more dynamic and their motions are much faster. Following the logic of the authors protein and membrane should move independent of each other. Still, membrane protein rotation (except rotation about the membrane normal) will be extremely restricted. It will almost never undergo flip-flop (meaning switching intra- and extracellular sides). So its rotation will be almost completely determined by its lipid environment. The same is true for peptides in micelles with the exception that the micelle will rotate freely and much faster than large membrane structures. Therefore the peptides will undergo flip-flop but only because the micelle undergoes flip-flop by rotating as a whole. Larger micelles will rotate slower and smaller ones will rotate faster. Any properties the

authors measured in their manuscript will mostly likely reflect that instead of any peptide motion by itself.

The second analysis uses the Stokes-Einstein equation. This is interesting but in my opinion the Stokes-Einstein equation needs some assumptions that are not really fulfilled by the simulations. The assumptions are that the sphere is rigid (which it clearly is not), that it even is a sphere (which micelles often are not) and that its surface is smooth (which it also very much is not). This explains why rotation of the micelles in the simulations is slower than predicted by the Stokes-Einstein equation. The interpretation of the authors that peptide and micelle rotation are independent would instead lead to faster rotation in the simulations because one would get additional rotation of the peptide on top of the rotation of the micelle as a whole. This should lead to shorter correlation times in the simulation (compared to the Stokes-Einstein equation) and not to longer ones. Therefore, this analysis seems too imprecise to be helpful and even contradicts the authors claims.

The third analysis I assume the authors refer to Fig. 7b. This is the only analysis that is at least partially convincing. I'm not sure however if the statistics can be done that way. I assume that each data point is considered independent when they are not really because the peptides rotate as a whole. I would like to see the same figure with micelle weight (sum of all SDS and peptide weights) on the x-axis (instead of helicity) using all simulations conducted with different micelle sizes and peptide dimers etc. This should give us a good indication of how much role micelle weight really plays for T1, T2 and hetNOE.

If my interpretation is true that micelle rotation mostly determines these experimental observables this negates the whole point of the manuscript. Instead one could make an article about what such experiments actually can and cannot show. Therefore, I would need to see convincing data that micelle rotation (not SDS rotation) and peptide rotation are independent before I can recommend publication of this article in Nat.Comm.Chem.

Reviewer #3 (Remarks to the Author):

Revisiting the manuscript "Probing the dynamic landscape of peptides in molecular assemblies by synergized NMR experiments and MD simulations" including the comments of the other reviewers I see the paper as a valuable contribution for the understanding of the dynamics of membrane peptides.

Considering the comments of all reviewers of the manuscript I can see three main concerns:

- 1) Doubts if SDS is a good membrane mimic.
- 2) Doubts if a micellar system is a good choice to characterize membrane peptide dynamics.

3) Interpretation of the dynamic parameters.

I did read the paper more as a method paper to elucidate the capability of solution NMR in combination with MD to investigate the dynamics of membrane peptides in micelles. Choice of a model system is always a compromise. Considering the ambitious task to connect MD with NMR relaxation the model system seems adequate to me. Of course solid state NMR with membranes would be an interesting approach. However, relaxation under MAS condition is an even more challenging task. (see for example publications of Paul Schanda)

In the updated version of the manuscript the authors improved the presentation of the interpretation of the dynamic parameters.

In total I think the manuscript a valuable contribution to the difficult topic of membrane peptide dynamics and should therefore be considered for publication.

Reviewer #1 (Remarks to the Author):

The manuscript "Probing the dynamic landscape of peptides in molecular assemblies by synergized NMR experiments and MD simulations" has been revised to address previous criticism. The authors acknowledged that their systems are not very good membrane mimics and changed the text and title accordingly. I think that this is a good decision since it emphasizes the more technical nature of this work.

AUTHOR REPLY:

We thank reviewer for constructive comments and for acknowledging the improvements in the revised version.

REVIEWER:

However, my main criticism was not really addressed in my opinion. I thought that the experimental results are mostly representative of micelle rotation instead of peptide dynamics and the authors have not convinced me otherwise. They did not conduct any other experiments, simulations or even analysis which would have been rather easy to prove either the authors or my view.

AUTHOR REPLY:

We apologize of not being sufficiently clear in our rebuttal letter enclosed with the revision, which has apparently left an impression that we have not conducted any further analyses on this topic. We actually performed additional analysis for the revision shown in Fig. S3, which gave further support for uncoupled motions of peptides and surfactant molecules. However, detailed discussion on this analysis was included in the reply to the comments by reviewer #3, who also proposed further analyses on this topic. We believe that this is the reason why reviewer #1 has missed this analysis.

To reply points by reviewer #3, we calculated correlation functions and timescales for the vectors between micelle's center of the mass and SDS sulfur atom, and reported these from different SDS molecules in figure S3 in the supplementary information. The related discussion was added in the "Peptide and surfactant rotations are uncoupled in micelles" section:

"... Overall rotation timescales of individual SDS molecules exhibit wide distribution in all systems with most molecules having timescales faster than approximately 4 ns. Slower timescales with small weights appear only in systems with peptides. Wide distribution of timescales for individual SDS molecules and the lack of common dominant timescales with peptides suggests that their overall motions are not concerted. On the other hand, appearance of slow timescales with small weights in systems with peptides indicate that few SDS molecules are attached to peptides such that they rotate partially together. Nevertheless, because the rotation of clear majority of SDS molecules are not coupled with peptides, we conclude that peptides rotate independently from detergent molecules in a viscous media formed by the micelle."

REVIEWER:

In Fig. 5 the authors have timescale data for each peptide but always for only one micelle size. It should have been easy to take the existing simulations for hMff(TA) with 40, 45, 50, and 60 SDS molecules and conduct the same analysis as in Fig. 5 for all four micelle sizes.

AUTHOR REPLY:

Spin relaxation times from micelles with different sizes in Fig. 3 indeed show small dependence on micelle size. This is related to slight slow down of peptide rotation with increasing micelle size:

**REVIEWER:**

If peptide and micelle rotation are independent the authors should get the same results for all of them since the peptide remains the same in all simulations and only the micelle size changes. I doubt that this would be the case though since Fig. 3a already shows that they lead to different predictions of experimental results. So judging from Figs. 1c, 3a and 3b, I still believe the almost all differences they see, originate from differences in micelle rotation.

AUTHOR REPLY:

We want to stress out that we do not argue that peptide rotation would be independent on the micelle. We argue that rotation of peptides and micelle molecules are uncoupled. We conclude that micelle forms a fluid-like environment for peptides:

“Our findings support a view of peptide dynamics within a detergent matrix in which peptides and detergent molecules do not rotate together as a rigid body in solvent. Rather, peptides rotate in a viscous medium composed of detergent micelle.”

Fig. 3 indicates that the peptide dynamics slightly depend on the size of this fluid-like environment. This is understandable because micelles are relatively small and do not encapsulate the peptide entirely. Therefore, in larger micelles a peptide interacts more with SDS molecules and less with the water than in smaller micelles, which leads to differences in peptide dynamics in micelles with different sizes.

On the other hand, yFis1(TA) and hMff(TA) have substantially larger T_1 times than eYqjD(TA) and Magainin2 in Fig. 4, even though all micelles have the same amount of SDS molecules (50). This indicates that differences between spin relaxation times cannot be explained only by the differences in micelle size.

It is also important to note that spin relaxation times are detected directly from peptides. Therefore, their differences will certainly originate from differences in peptide motions. The question is how peptide dynamics is affected by the micelle environment. Our view is that micelle molecules form a fluid-like viscous environment for a peptide.

REVIEWER:

The authors try to negate this interpretation by showing that peptide and micelle rotations are uncoupled. However, I think in particular the first analysis about the correlation times is very misleading and doesn't tell us anything about the question.

AUTHOR REPLY:

We agree with the reviewer that the correlation time analysis alone may not be fully conclusive, as also pointed out by the reviewer #3. In the original version of manuscript, we tackled this issue by performing the independent analysis using Stokes-Einstein equation, which also indicated that a micelle cannot be considered as a rigid body but peptide rather rotates in a viscous fluid-like environment formed by micelle molecules. In Fig. S3 in the revised version of the manuscript, we performed additional analysis on overall rotations of SDS molecules, which also supported the conclusion that peptide and surfactant molecule rotations are uncoupled. Furthermore, we have now added analyses on SDS rotation after removing the overall rotation of a peptide from simulations of monomeric GWALP peptides, which gives further support for our conclusions (see Fig. S4 and replies to comments below).

REVIEWER:

Of course peptide and surfactant rotations are uncoupled. This is absolutely expected. The molecular weight of SDS is that of about 3 alanines. So all peptides are much heavier than a SDS molecule. Also internal flexibility of SDS is much higher than that of alpha-helices. One would have to compare peptide motion to micelle motion as a whole and not to SDS motion.

AUTHOR REPLY:

This is probably one of the key points where our view diverges from the reviewer's. We argue in the manuscript that peptide and surfactant (not the whole micelle) rotations are uncoupled, but this statement is far from trivial. Without the data presented in our manuscript, it could be possible that a peptide and all surfactant molecules in a micelle would share common rotational modes that could be interpreted as overall micelle rotation. If this coupling would be strong enough, micelle and peptide rotations could be approximated as a rigid body using Stokes-Einstein equation.

We observe that SDS molecules form fluid-like micelles and exhibit liquid like motions with higher flexibility than that of alpha-helices. However, this is also not trivial. For example, Amber force field predicts much more ordered micelles, but the results do not agree with experiments in Fig. 2. Amber force field would probably give different conclusion also for peptide-surfactant coupling. However, based on our direct comparison with experiments, we show that fluid-like micelles predicted by CHARMM parameters are in good agreement with the experimental NMR data.

Reviewer is suggesting here and in their first comments to analyze micelle motion as a whole. However, it is not possible to define such motion because micelles are fluid-like objects where molecules move independently with respect to each others. Nevertheless, for peptides with rigid structures, it is possible to remove the peptide rotation from the trajectories and then analyze surfactant rotation. We have now performed such analysis for monomeric GWALP peptides in micelles with different number of surfactants. The results are added in Fig. S4 in the supplementary information, and a paragraph is added in "*Peptide and surfactant rotations are uncoupled in micelles*" section:

“Our results suggest that peptide and surfactant rotations do not share common timescales in micelles, and therefore a common timescale that would describe the rotation of a micelle as a whole cannot be defined. Nevertheless, for peptides with rigid structures, it is possible to remove the peptide rotation from the trajectories and then analyze surfactant rotation. To further demonstrate uncoupling of peptide and surfactant rotations in micelles, we removed peptide rotation from trajectories of GWALP monomers in micelles with different amount of SDS molecules, and then compared SDS overall rotation from these trajectories with the original trajectories. Results in Fig. S4 show similar SDS rotations in trajectories with and without peptide rotation removal, providing further evidence that peptide and surfactant rotations are uncoupled in micelles.”

REVIEWER:

Imagine a membrane protein inside a membrane. Measuring any dynamic property of the lipids and the protein and comparing them, one will find the lipids are way more dynamic and their motions are much faster. Following the logic of the authors protein and membrane should move independent of each other. Still, membrane protein rotation (except rotation about the membrane normal) will be extremely restricted. It will almost never undergo flip-flop (meaning switching intra- and extracellular sides). So its rotation will be almost completely be determined by its lipid environment.

AUTHOR REPLY:

In similar analyses for membrane systems, the rotational correlation functions of, e.g. C-H bonds, first relax with nanosecond timescales to the plateau value, which is the square of the order parameter of the bond which can be measured using dipolar or quadrupolar coupling. The correlation function would then plateau to zero with millisecond timescales due to the magic angle spinning and/or diffusion of lipids between membranes in different orientations. These timescales are too long for atomistic resolution MD simulations. Therefore, similar membrane studies are focused on dynamics that happens before correlation functions relaxing to the square of order parameter. See for example, Ferreira et al. [*J. Chem. Phys.* 142, 044905 (2015), <https://doi.org/10.1063/1.4906274>].

If we were able to simulate overall rotation and/or molecule diffusion between planes with different orientations in multi-lamellar sample, then we could perform similar analyses also for membrane systems to see if there would be common timescales in these slow motions. However, it is not feasible to run millisecond long MD simulations with atomistic resolution.

REVIEWER:

The same is true for peptides in micelles with the exception that the micelle will rotate freely and much faster than large membrane structures. Therefore the peptides will undergo flip-flop but only because the micelle undergoes flip-flop by rotating as a whole.

AUTHOR REPLY:

Molecules in micelles indeed rotate faster and their correlation functions decay to zero in the timescales that are accessible with MD simulations. However, “flip-flop” event is not well defined in a micelle because all micelle molecules rotates randomly without any symmetry axis along which the “flip-flop” could be defined.

Figure below illustrates different behaviour of surfactants and lipids in a micelle and membrane in simulations containing eElab(TA) peptides. Time evolution of individual surfactant and lipid molecules are illustrated by showing overlaid snapshots between 1ns. In both illustrations, the orientation of peptide is fixed. In a micelle, SDS molecules rotate freely with respect to peptide orientation, while in membrane their motion is fixed with respect to the membrane. Panel a) also illustrates that removing peptide rotation does not have substantial effect on SDS rotation.

**REVIEWER:**

Larger micelles will rotate slower and smaller ones will rotate faster. Any properties the authors measured in their manuscript will mostly likely reflect that instead of any peptide motion by itself.

AUTHOR REPLY:

Peptides in larger micelles indeed rotate slightly slower than in smaller micelles as already discussed above. However, timescales of SDS molecules in Fig. 5 b) are similar to all systems despite the differences in micelle size and peptide rotation. This applies also to GWALP simulations where the micelle is substantially larger and peptide rotation is slower. This again supports the view that SDS molecules rotate independently on peptides, and that the micelle rotation cannot be well defined as a whole.

Note that when replying to this comment we realized that GWALP results were missing from Fig. 5 b) in previous versions of the manuscript. That data is now added in the figure.

REVIEWER:

The second analysis uses the Stokes-Einstein equation. This is interesting but in my opinion

the Stokes-Einstein equation needs some assumptions that are not really fulfilled by the simulations. The assumptions are that the sphere is rigid (which it clearly is not), that it even is a sphere (which micelles often are not) and that its surface is smooth (which it also very much is not). This explains why rotation of the micelles in the simulations is slower than predicted by the Stokes-Einstein equation.

AUTHOR REPLY:

If a peptide would rotate together with the surfactant molecules, it should be possible to approximately describe the peptide rotation motion with Stokes-Einstein equation. It is true that irregularities in micelle shape and surface will affect the exact numerical values in the Stokes-Einstein equation. For this reason we have discussed on the range of numerical values that would be required to explain observed dynamics with Stokes-Einstein equation in the section *“Peptide and surfactant rotations are uncoupled in micelles”*:

“For example, in the case of the hMff system in 50 SDS molecules, the gyromagnetic radius calculated from the simulations is 1.6 nm. However, to obtain the rotational timescales observed in simulations from the Stokes-Einstein equation, the radius of the micelle would have to be 3.0 nm, which is almost twice as large as the value from the simulations. On the other hand, the viscosity value in Stokes-Einstein equation should be 7.5 mPa s to obtain the same dynamics as observed in simulations, which is approximately ten times larger than the viscosity of water at 310 K (approximately 0.69 mPa s).”

The Stokes-Einstein predictions are too far from the observed dynamics that they could be explained by small inaccuracies in the assumptions about spherical shape or surface roughness.

We agree that this discrepancy is because the micelle dynamics cannot be described as a rigid body, as we write in section *“Peptide and surfactant rotations are uncoupled in micelles”*:

“Because peptides rotate independently from detergents in a viscous media formed by the micelle, the rotation of the peptide-micelle complex cannot be described by the Stokes-Einstein equation that assumes that peptides and detergents rotate together as a spherical rigid body.”

REVIEWER:

The interpretation of the authors that peptide and micelle rotation are independent would instead lead to faster rotation in the simulations because one would get additional rotation of the peptide on top of the rotation of the micelle as a whole. This should lead to shorter correlation times in the simulation (compared to the Stokes-Einstein equation) and not to longer ones. Therefore, this analysis seems too imprecise to be helpful and even contradicts the authors claims.

AUTHOR REPLY:

Our interpretation is that a peptide in a micelle rotates in viscous media formed by surfactant molecules. Because viscosity of such environment is higher than for water, the rotation of a peptide is slower than it would be in a rigid micelle in water. Therefore, our results do not contradict with our conclusion that peptide rotates in a viscous media formed by surfactant molecules.

REVIEWER:

The third analysis I assume the authors refer to Fig. 7b. This is the only analysis that is at least partially convincing. I'm not sure however if the statistics can be done that way. I assume that each data point is considered independent when they are not really because the peptides rotate as a whole. I would like to see the same figure with micelle weight (sum of all SDS and peptide weights) on the x-axis (instead of helicity) using all simulations conducted with different micelle sizes and peptide dimers etc. This should give us a good indication of how

much role micelle weight really plays for T1, T2 and hetNOE.

AUTHOR REPLY:

With the third analysis we referred to the analysis of correlation functions and timescales for the vectors between micelle's center of the mass and SDS sulfur atom shown in figure S3. We apologize again that we were not more clear in this in our first reply.

The point of Fig. 7B is to investigate how strongly spin relaxation times correlate with helicity as discussed in section “*Correlations between spin relaxation times and peptide secondary structure in micelles*”. This is a different topic. Nevertheless, below we show the plot requested by the reviewer. This plot is in agreement with discussion related to Fig. 3. in section “*MD simulations predicting spin relaxation times of peptide-micelle complexes.*”:

“*The results in Fig 3 a) show systematic but weak dependence of T_1 on micelle sizes, while T_2 and hetNOE spin relaxation times from differently sized micelles are mostly within the error bars.*”

REVIEWER:

If my interpretation is true that micelle rotation mostly determines these experimental observables this negates the whole point of the manuscript. Instead one could make an article about what such experiments actually can and cannot show. Therefore, I would need to see convincing data that micelle rotation (not SDS rotation) and peptide rotation are independent before I can recommend publication of this article in Nat.Comm.Chem.

AUTHOR REPLY:

Reviewer's criticism concerns this conclusion: “*Our findings support a view of peptide dynamics within a detergent matrix in which peptides and detergent molecules do not rotate together as a rigid body in solvent. Rather, peptides rotate in a viscous medium composed of detergent micelle.*” We originally presented two analyses that supported this conclusion (bond vector correlations and Stokes-Einstein equation). In the first revision, we added the analysis about rotation of vectors between micelle center and phosphorus atoms (see Fig. S3). We have now added further evidence in Fig. S4 where we analyze SDS rotations from systems with peptide rotation removed and compare the results to the original trajectories. Also visual inspection of trajectories support this interpretation.

We summarize here the main points in which reviewer's view is different to ours. We hope that these will help to better understand the discussion.

- 1) Reviewer is asking evidence that micelle and peptide rotation are independent. However, we do not argue that peptide rotation would be independent of a micelle. We argue that a peptide in micelle does not rotate together with surfactant molecules, but it rotates in fluid-like viscous media composed of detergent molecules.
- 2) Reviewer is suggesting to analyse the micelle rotation as a whole. However, such rotation is not well defined because micelles are fluid-like objects and surfactant molecules rotate independently with respect to each others.
- 3) Reviewer argues that measured spin relaxation times would be representative of micelle rotation, not protein rotation. However, ^{15}N spin relaxation times are detected directly from peptides. Therefore, they will certainly originate from peptide motions. The question is then how peptide dynamics is affected by the micelle environment. Our view is that micelle molecules form a fluid-like viscous environment for a peptide.

We also summarize here the presented evidence for our conclusion *“Our findings support a view of peptide dynamics within a detergent matrix in which peptides and detergent molecules do not rotate together as a rigid body in solvent. Rather, peptides rotate in a viscous medium composed of detergent micelle.”*:

- 1) Peptides and SDS molecules do not have common timescales in Fig. 5.
- 2) Peptide rotation cannot be described using Stokes-Einstein equation by assuming that peptide and surfactant molecules rotate together as a rigid body (Fig. 6).
- 3) Overall rotation timescales of almost all SDS molecules are substantially faster than peptides (Fig. S3).
- 4) SDS rotation is essentially similar in original trajectories and in trajectories from where the peptide rotation has been removed (Fig. S4).

Furthermore, conclusion about peptide and surfactant molecule uncoupling is not the only results in our manuscript. Therefore, we find the reviewer's statement about negating the whole point of the manuscript to be slightly exaggerating. For example, we discuss in the manuscript also on which properties experimental data is sensitive to, which helps to understand what these experiments can and cannot show: *“We found significant correlations of helical propensities of peptide residues with large hetNOE and low T_2 values, particularly when compared with the other residues within the same peptide. On the other hand, T_1 values mainly correlated with the overall rotation of peptides.”*

We hope that these clarifications will help to understand our arguments better.

Reviewer #3 (Remarks to the Author):

Revisiting the manuscript “Probing the dynamic landscape of peptides in molecular assemblies by synergized NMR experiments and MD simulations” including the comments of the other reviewers I see the paper as a valuable contribution for the understanding of the dynamics of membrane peptides.

Considering the comments of all reviewers of the manuscript I can see three main concerns:

- 1) **Doubts if SDS is a good membrane mimic.**
- 2) **Doubts if a micellar system is a good choice to characterize membrane peptide dynamics.**
- 3) **Interpretation of the dynamic parameters.**

I did read the paper more as a method paper to elucidate the capability of solution NMR in combination with MD to investigate the dynamics of membrane peptides in micelles. Choice

of a model system is always a compromise. Considering the ambitious task to connect MD with NMR relaxation the model system seems adequate to me. Of course solid state NMR with membranes would be an interesting approach. However, relaxation under MAS condition an even more challenging task. (see for example publications of Paul Schanda)

In the updated version of the manuscript the authors improved the presentation of the interpretation of the dynamic parameters.

In total I think the manuscript a valuable contribution to the difficult topic of membrane peptide dynamics and should therefore be considered for publication.

AUTHOR REPLY:

We thank reviewer for positive comments.

REVIEWERS' COMMENTS:

Reviewer #1 (Remarks to the Author):

The manuscript "Probing the dynamic landscape of peptides in molecular assemblies by synergized NMR experiments and MD simulations" has been revised on more time to address previous criticism.

I want to thank the authors for the effort they put into the rebuttal letter. In fact I did miss Figure S3 the last time. Also the new Figure S4 is very helpful and the figure in which the correlation between T1, T2, and hetNOE and molecular weight is shown is very interesting and in my opinion should be added to the SI.

In summary the new data is very convincing that SDS and the peptides in fact do rotate independently. However, I would like the influence of micelle size on peptide rotation and hence in particular T1 to be discussed more prominently. This is important for the practical application since micelle size would have to be controlled very precisely for good agreement between simulation and experiment.

Therefore, I can now recommend publication of this article in Nat.Comm.Chem. after some minor revisions.

Reviewer #1 (Remarks to the Author):

The manuscript "Probing the dynamic landscape of peptides in molecular assemblies by synergized NMR experiments and MD simulations" has been revised on more time to address previous criticism.

I want to thank the authors for the effort they put into the rebuttal letter. In fact I did miss Figure S3 the last time. Also the new Figure S4 is very helpful and the figure in which the correlation between T₁, T₂, and hetNOE and molecular weight is shown is very interesting and in my opinion should be added to the SI.

AUTHOR REPLY:

We thank reviewer for positive comments.

The mentioned figure showing correlation between T₁, T₂, and hetNOE and molecular weight is now added in the supplementary information (Supplementary Figure 5). Text referring to the new figure is now added in page 9 in the main text: *"On the other hand, T₁ has strong positive, T₂ mild negative, and netNOE very mild positive correlation with the micelle molecular weight in Supplementary Figure 5, in line with conclusions in above sections."*

REVIEWER:

In summary the new data is very convincing that SDS and the peptides in fact do rotate independently. However, I would like the influence of micelle size on peptide rotation and hence in particular T₁ to be discussed more prominently. This is important for the practical application since micelle size would have to be controlled very precisely for good agreement between simulation and experiment.

AUTHOR REPLY:

We are happy to see that we have now found an agreement with the reviewer.

We agree that influence of micelle size on peptide rotation is important for reaching good agreement between simulation and experiment. This topic is discussed quite extensively in section *"MD simulations predicting spin relaxation times of peptide-micelle complexes"*. Figure 3 shows how spin relaxation times depend on micelle size for selected systems. This data is used to find optimal micelle sizes that reproduce the experimental spin relaxation times and enable interpretation of experiments. We do comment results on individual peptides, for example for hMff(TA), we write on page 9: *"The results in Fig 3 a) show systematic but weak dependence of T₁ on micelle sizes, while T₂ and hetNOE spin relaxation times from differently sized micelles are mostly within the error bars."* However, for more general discussion about micelle size, we should simulate more peptides with different sizes, which would require substantial computational resources, as well as careful analyses and discussion. Therefore, we have opted to focus here on finding correct micelle sizes for the interpretation of experiments and leave more general discussions on effect of micelle size on dynamics for future studies.

Therefore, I can now recommend publication of this article in Nat.Comm.Chem. after some minor revisions.

AUTHOR REPLY:

We thank reviewer for the positive recommendation.